# Seeing Symbols, Missing Structure: A Real-World Handwritten Mathematical Expression Recognition Benchmark for Large Models

**Sheng Jiang** [1 2 3]   **Lin Zhu** [1 2 3]   **Runrui Li** [1 2 3]   **Mei Wang** [1 2 3]   **Qiannan Zhu** [1 2 3]   **Yaoyao Zhong** [1 2 3]
**Hua Huang** [1 2 3]

## Abstract

Handwritten mathematical expression recognition (HMER) remains challenging in real-world educational scenarios, even with recent advances in large vision-language models. While these models often achieve high accuracy in local symbol transcription, their reliability in capturing two-dimensional mathematical structure under realistic handwritten conditions is still poorly understood. We introduce a real-world handwritten benchmark covering 13 categories of structurally complex expressions with authentic writing artifacts. Evaluations on large models reveal a clear performance degradation as structural complexity increases, even when symbol-level accuracy is high. Most failures arise from structural mis-parsing and context-dependent symbol role confusion rather than pure visual perception errors. To mitigate this issue, we propose a training-free, schema-anchored structure-aware inference framework that decomposes recognition into schema identification, schema-constrained transcription, and context-driven disambiguation. Our method improves the ExpRate from 11.63% to 24.52% on Qwen-8B and generalizes well across multiple large models. Our benchmark provides a realistic evaluation for large models on handwritten mathematics, and our framework offers an effective and interpretable solution to structure-related failures in real-world HMER. Code and data are available at: https://github.com/BNU-ERC-ITEA/HMER-Bench.git.

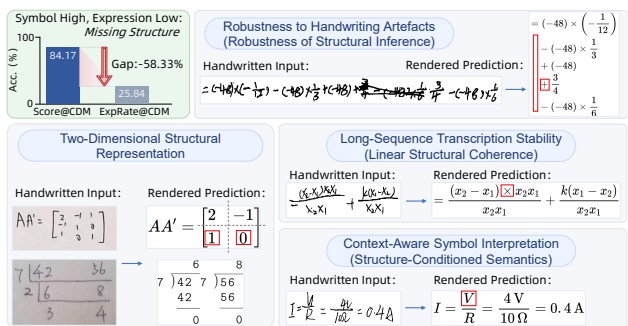

*Figure 1.* **Reproducible failure modes of VLMs on HMER in real educational scenarios.** Symbol/Expression Accuracy are Score@CDM/ExpRate@CDM on HMER-Bench; red boxes highlight the erroneous positions in the prediction.

---

[1] School of Artificial Intelligence, Beijing Normal University, Beijing 100875, China [2] Beijing Key Laboratory of Artificial Intelligence for Education, Beijing 100875, China [3] Engineering Research Center of Intelligent Technology and Educational Application, Ministry of Education, Beijing 100875, China. Correspondence to: Lin Zhu <linzhu@bnu.edu.cn>.

*Proceedings of the 43rd International Conference on Machine Learning*, Seoul, South Korea. PMLR 306, 2026. Copyright 2026 by the author(s).

## 1. Introduction

Handwritten Mathematical Expression Recognition (HMER) is a fundamental problem in document understanding and scientific content digitization, with direct applications in educational assessment, automatic grading, and knowledge extraction from handwritten materials. Unlike plain text recognition, HMER requires recovering not only symbol identities but also the latent two-dimensional (2D) structure (Firdaus & Vaidehi, 2019) that governs mathematical semantics, including spatial alignment, hierarchical grouping, and cross-line dependencies.

Recent advances in encoder–decoder architectures and large vision–language models (VLMs) have substantially improved local symbol transcription accuracy (Yang et al., 2025b). However, in real educational scenarios, we observe a persistent gap between symbol-level correctness and expression-level validity. As illustrated in Fig. 1, models frequently produce outputs that contain mostly correct symbols but violate global structural constraints, such as mistaking single-line expressions as multi-line ones, confusing short division with long division, miscounting matrix size, or producing duplicated symbols, among others. These failures indicate that the dominant bottleneck of modern HMER systems lies not in visual perception, but in *structural reasoning and global layout modeling*.

We further identify three representative manifestations of this structural bottleneck. First, models struggle to maintain consistent 2D layouts as expressions become longer and more nested, leading to omissions, duplications, or disrupted alignment. Second, symbol meanings are often misinterpreted when their structural roles are ambiguous, resulting in contextually invalid but visually plausible predictions. Third, structural inference becomes highly unstable under authentic handwriting artefacts such as overwriting, erasures, and cancellation marks. Notably, these errors often emerge during the inference stage even when the majority of symbols are correctly detected, suggesting that end-to-end sequence decoding without explicit structural control is fundamentally fragile.

Despite this limitation, existing benchmarks and evaluation protocols are insufficient to characterize structural failures in a systematic manner. Popular datasets such as CROHME (Mouchere et al., 2014; Mouchère et al., 2016; Mahdavi et al., 2019) and HME100K (Yuan et al., 2022) primarily focus on single-line expressions, while multi-line datasets provide limited coverage of row–column alignment, primary-school layouts (e.g., vertical arithmetic and short division), and real-world writing artefacts. As a result, current evaluations fail to answer a critical question: *which structural capabilities are missing in current large models, and under what structural conditions do they collapse?*

To address this gap, we introduce **HMER-Bench**, a real-world handwritten mathematical expression benchmark designed from a structure-centric perspective. Our benchmark is constructed from authentic handwritten examination and homework data collected under natural writing conditions, capturing the structural complexity of educational scenarios and enabling reliable evaluation of structural reasoning in handwritten mathematical expression recognition. HMER-Bench covers 13 fine-grained categories organized into four diagnostic dimensions, targeting linear structural coherence, two-dimensional layout representation, structure-conditioned symbol semantics, and robustness of structural inference.

Based on the diagnostic findings on HMER-Bench, we further propose a *training-free schema-anchored structure-aware reasoning framework*. By introducing a latent structural variable to represent the global layout type and enforcing schema-consistent decoding, our approach explicitly decomposes recognition into structural schema inference, structure-constrained transcription, and context-driven symbol disambiguation. This design substantially reduces globally inconsistent predictions without requiring additional training data or parameter updates.

In summary, our contributions are threefold:

(1) We propose **HMER-Bench**, the first benchmark that systematically evaluates structural reasoning capabilities of HMER systems under realistic educational conditions.

(2) Through large-scale evaluation, we demonstrate that insufficient structural modeling, rather than symbol recognition, constitutes the primary performance bottleneck of modern VLM-based HMER systems.

(3) We introduce a schema-anchored structure-aware inference framework that significantly improves recognition stability, interpretability, and accuracy across complex 2D layouts. The benchmark will be made publicly available, establishing a principled foundation for structure-centric handwritten mathematical expression recognition.

## 2. Related Works

**HMER Methods.** Most HMER systems adopt encoder–decoder architectures. WAP (Zhang et al., 2017) introduces attention-based end-to-end recognition, and Dense-WAP (Zhang et al., 2018) strengthens feature representation using DenseNet. To better handle long sequences and complex layouts, BTTR (Zhao et al., 2021) employs bidirectional Transformer decoders, while CoMER (Zhao & Gao, 2022) integrates coverage attention to reduce omissions and repetitions. Beyond sequential modeling, CAN (Li et al., 2022) enforces global constraints via symbol counting, ICAL (Zhu et al., 2024) enhances implicit LaTeX structure awareness, and SAN (Yuan et al., 2022), TAMER (Zhu et al., 2025), and PosFormer (Guan et al., 2024) explicitly incorporate syntactic trees or relative positional modeling to improve structural consistency.

Recently, VLMs have been introduced for expression and document recognition. With large-scale and diverse pre-training data and strong generalization ability, these models have demonstrated competitive performance on HMER tasks, surpassing traditional approaches in many settings. HunyuanOCR (Team et al., 2025), PaddleOCR-VL (Cui et al., 2025), and MonkeyOCR (Li et al., 2025c) exploit instruction-driven generation and structure-aware decoding for complex layouts. For handwritten expressions, HiE-VL (Guo et al., 2025b) and VLPG (Guo et al., 2025a) adopt hierarchical or graph-based adapters, while Uni-MuMER (Li et al., 2026) performs multi-task fine-tuning with tree inference and symbol disambiguation.

Despite these advances, VLM-based approaches still suffer substantial performance degradation on expressions with complex two-dimensional structures, indicating limited global structural reasoning capability.

**Datasets and OCR Benchmarks.** Regarding data and evaluation, single-row benchmarks such as CROHME and HME100K are representative, whilst multi-row datasets like $M^2E$ (Yang et al., 2023b) focus on row segmenta-

*Table 1.* **Dataset comparison (✓/△/✗denote full/partial/none support).** Edu: real-world educational source; 2D: diverse 2D layouts; Align: row/column alignment; P-S: primary-school layouts (vertical arithmetic, short division) with unified annotation; Noise: artifacts/noise coverage; Tax: fine-grained diagnostic taxonomy.

| Dataset | Edu | 2D | Align | PriSch | Noise | Tax |
|---|---|---|---|---|---|---|
| CROHME | ✗ | ✗ | ✗ | ✗ | ✗ | ✗ |
| HME100K | ✓ | ✗ | ✗ | ✗ | △ | △ |
| $M^2E$ | ✓ | △ | ✗ | ✓ | △ | ✗ |
| MLHME-38K | ✓ | △ | ✗ | ✗ | △ | ✗ |
| MathWriting | ✗ | △ | ✗ | ✗ | ✗ | ✗ |
| HMER-Bench | ✓ | ✓ | ✓ | ✓ | ✓ | ✓ |

tion but inadequately model inter-row semantic associations and column alignment; MathWriting (Gervais et al., 2025) leans towards synthetic data and online scenarios, while MLHME-38K features multi-row equations. OCR-Bench (Liu et al., 2024)/OCRBench v2 (Fu et al., 2026) incorporates HME100K and some multi-row datasets into the HMER sub-task, yet lacks fine-grained classification criteria to diagnose structural capabilities. FERMAT (Nath et al., 2025) benchmarks VLMs for grading handwritten math solutions via error detection/localization/correction, but it is not designed to diagnose fine-grained 2D expression layouts or LaTeX transcription failures in complex structures. As shown in Table 1, HMER-Bench provides the most systematic coverage across Edu/2D/Align/PriSch/Noise/Tax, while existing benchmarks typically cover only a subset.

**Structural Reasoning in Large Models.** Although VLMs achieve strong semantic recognition, they remain brittle in 2D spatial/structural reasoning. What'sUp (Kamath et al., 2023), SpatialEval (Wang et al., 2024), and Mind the Gap (Stogiannidis et al., 2025) report near-chance performance under controlled 2D relational changes. This limitation is partly attributed to flattening images into one-dimensional token sequences with RoPE-1D (Alam et al., 2026), while introducing 2D positional encodings (Li et al., 2025a) improves structural fidelity. Recent reasoning-enhancement methods, such as chain-of-thought prompting and region-guided cues (Yang et al., 2023a), provide partial gains but may amplify local errors into globally inconsistent structures. Uni-MuMER further introduces Tree-CoT supervision during training, yet achieving stable global consistency for lattice layouts (e.g., matrices) remains difficult.

Despite these efforts, structure-aware reasoning for real-world HMER remains largely unexplored. Existing studies mainly focus on generic spatial relations or training-time supervision. In contrast, our work proposes a test-time, structure-aware reasoning framework that explicitly separates layout inference from symbol transcription and semantic disambiguation, improving structural consistency without additional training or parameter updates.

# 3. Handwritten Mathematical Expression Recognition Benchmark

Our benchmark is constructed from authentic handwritten examination data collected under natural writing conditions, capturing the structural complexity of real educational scenarios and enabling reliable evaluation of structural reasoning in handwritten mathematical expression recognition.

## 3.1. Diagnostic Dimensions and Category Mapping

We identify recurring failure patterns in authentic educational settings and abstract model bottlenecks into four diagnostic dimensions. Although these dimensions capture different surface-level phenomena, they are all rooted in a common cause: the model's ability to correctly infer and maintain the global structure of mathematical expressions.

In particular, we view long-sequence stability as *linear structural coherence*, symbol interpretation as *semantic grounding conditioned on structure*, and robustness to handwriting artefacts as the *stability of structural inference under perturbations*. Under this perspective, two-dimensional layout modeling serves as the core structural capability, while the remaining dimensions characterize how structure is preserved, utilized, and stabilized during decoding.

Based on this framework, we construct **HMER-Bench** with 13 categories, enabling recognition results to be explicitly attributed to specific structural capability shortcomings.

**D1: Long-Sequence Transcription Stability (Linear Structural Coherence).** This dimension evaluates linear structural coherence, i.e., whether the model can preserve a globally consistent one-dimensional structure as expression length increases. Structural degradation in this setting typically manifests as symbol omission, duplication, or reordering. It includes *Short Single-Line Expressions (SSE)* and *Long Single-Line Expressions (LSE)*, which isolate sequence-level structural consistency from higher-order spatial layout complexity.

**D2: Two-Dimensional Structural Representation.** This dimension evaluates the model's ability to reconstruct hierarchical layouts and spatial alignment constraints that cannot be captured by linear order alone.

It comprises six categories: *Single-line Derivations (SLD)*, *Multi-line Derivations (MLD)*, *Short Division Expressions (SDE)*, *Vertical Arithmetic Expressions (VAE)*, *Systems of Equations (SOE)*, and *Matrices and Determinants (MAD)*. SLD and MLD test segmentation and reading-order inference across derivation steps. SDE and VAE emphasize column-wise alignment constraints induced by procedural layouts. SOE and MAD require consistent reconstruction of implicit row–column grids and block-level alignment in multi-line structures. Collectively, these categories char-

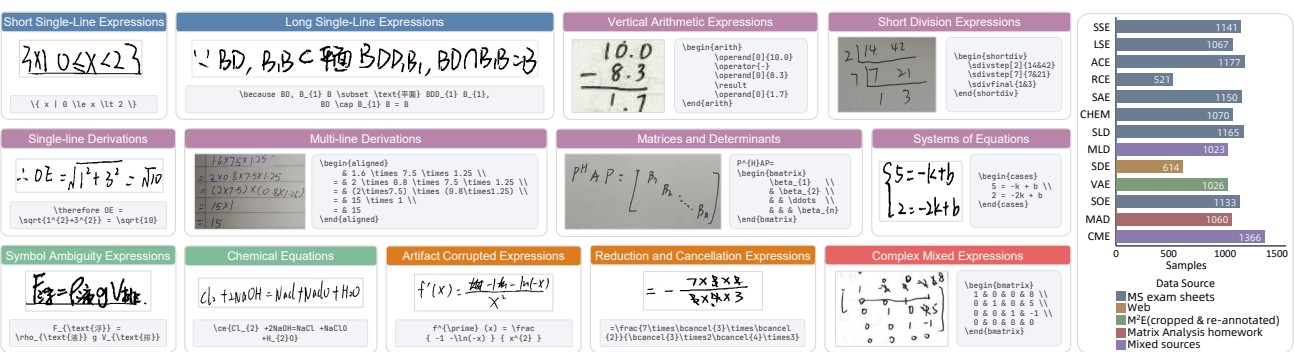

*Figure 2.* **HMER-Bench overview**: representative samples with ground-truth annotations, and per-category statistics (count, source).

acterize a model's capacity to infer latent 2D layouts and enforce global geometric consistency.

**D3: Context-Aware Symbol Interpretation (Structure-Conditioned Semantics).** This dimension evaluates semantic grounding conditioned on structure, where symbol identity and functional role must be inferred jointly from visual appearance and the inferred global layout. It includes *Symbol Ambiguity Expressions (SAE)* and *Chemical Equations (CHEM)*. SAE focuses on resolving homographs (e.g., "l" vs. "1", "O" vs. "0") using mathematical syntax and structural context, while CHEM requires discipline-specific structural rules and short-range dependency modeling.

**D4: Robustness to Handwriting Artefacts (Robustness of Structural Inference).** This dimension evaluates whether structural inference remains stable under perturbations introduced by authentic handwriting artefacts such as overwriting, smearing, or crossing out. It contains *Artifact Corrupted Expressions (ACE)* and *Reduction and Cancellation Expressions (RCE)*. ACE targets non-semantic noise that should be ignored during parsing. RCE involves semantically meaningful artefacts (e.g., strikethrough-based cancellation), requiring the model to distinguish structural operations from noise and correctly reflect them in the final representation.

Beyond isolated capabilities, *Complex Mixed Expressions (CME)* combine multiple structural challenges within a single expression, such as nested matrices with derivations or arithmetic with cancellation marks. This category approximates the compositional complexity of real exam scripts and serves as a holistic stress test for structural reasoning. Fig. 2 presents representative examples from all categories with their corresponding ground-truth annotations. Together, these dimensions form a structure-centric diagnostic taxonomy that links observable transcription failures to their underlying causes in layout modeling, alignment inference, semantic grounding, and structural robustness.

### 3.2. Dataset Statistics and Annotation

The HMER-Bench comprises 13,513 handwritten mathematical expression samples. Fig. 2 details the sample counts and data sources for each category. All samples underwent manual selection and annotation, with multiple rounds of verification to ensure data quality and annotation consistency. We provide `arithstudio.sty` for annotation and stable rendering of VAE and SDE, for which unified annotation standards are still lacking. More details can be found in the supplementary materials.

## 4. Method

### 4.1. Problem Reformulation

Traditional HMER methods typically formalise the task as maximising the posterior probability $P(Y \mid X)$ (Zhang et al., 2025; Guan et al., 2024), where $X$ denotes the input image and $Y$ represents the target LaTeX sequence. This sequential formulation implicitly assumes left-to-right linear decoding. However, complex handwritten mathematical expression exhibit diverse two-dimensional structures, with the semantic role of visual symbols being highly dependent on their global structural context. For instance, a horizontal line segment may denote a minus sign, a fraction bar, or a chemical bond. If symbol-level decoding proceeds without first establishing the global structure, the model becomes highly susceptible to local ambiguities. These ambiguities are then amplified during subsequent decoding stages, ultimately leading to expression recognition failure.

To explicitly model this structural dependency, we reformulate the HMER task as a structurally conditioned transcription problem. We introduce a latent structural variable $Z$, termed the structural schema, to characterise the layout type of the expression, and decompose the generative process as:

$$P(Y \mid X) = \sum_{z \in \mathcal{Z}} P(Y \mid X, z) \cdot P(z \mid X), \quad (1)$$

where $\mathcal{Z}$ denotes the set of predefined structural schemas, $P(z \mid X)$ corresponds to inferring the global structural

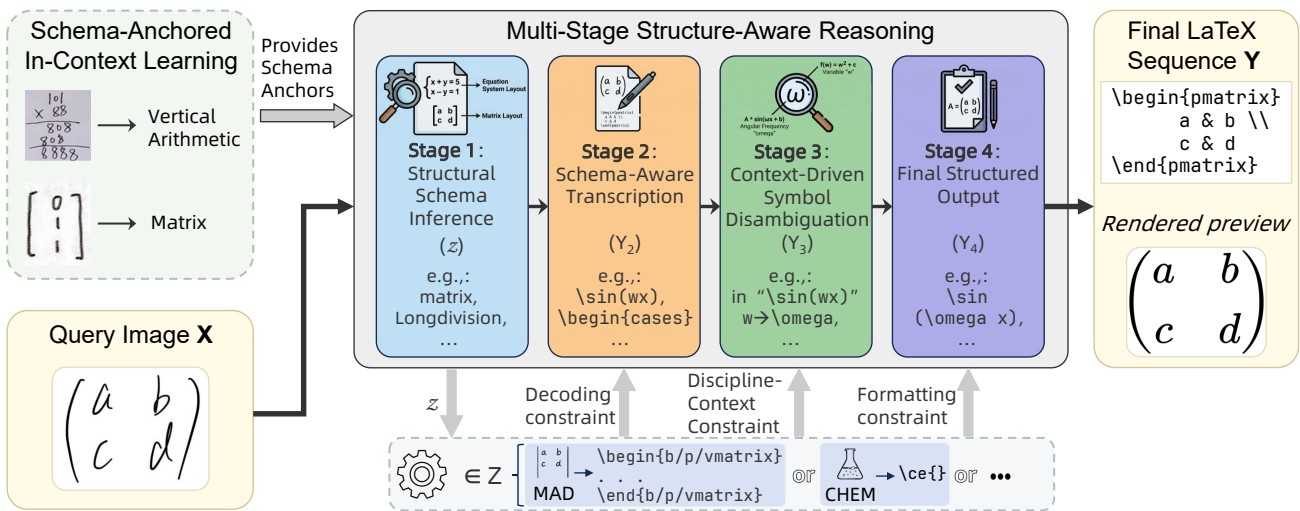

*Figure 3.* Overview of Schema-Anchored Structure-Aware Reasoning (SASR).

category of the expression from the image (matrix, vertical, etc.), while $P(Y \mid X, z)$ generates the specific LaTeX expression under the corresponding structural constraints. This decomposition explicitly distinguishes between the two subproblems of two-dimensional layout understanding and symbolic-level generation, thereby effectively constraining the output space and reducing ambiguity.

### 4.2. Multi-Stage Structure-Aware Reasoning (MSR)

Based on the aforementioned decomposition, we employ a multi-stage reasoning process to approximate inference of the latent structural variable $Z$ and generation under structural constraints (Fig. 3). Specifically, the entire decoding procedure is divided into four stages: (1) structural schema inference, (2) structure-aware transcription, (3) context-driven symbol disambiguation, and (4) final structured output. Formally, let $z$ denote the structural schema, $Y_2$ the initial LaTeX transcription result, $Y_3$ the disambiguation signal, and $Y_4$ the final output. The overall generation process can then be expressed as follows:

$$P(Y_4, Y_3, Y_2, z \mid X) = P(z \mid X)P(Y_2 \mid X, z)$$
$$P(Y_3 \mid X, z, Y_2)P(Y_4 \mid X, z, Y_2, Y_3). \quad (2)$$

The structural categories output in the first stage directly determine the permissible structural syntax for subsequent processing, thereby establishing 'two-dimensional structural modelling' as an explicit prerequisite. The third stage does not involve mathematical computation but rather performs semantic disambiguation of contextually ambiguous symbols to prevent errors arising from structurally correct but misinterpreted symbol meanings.

By conditioning transcription on the inferred structural schema, MSR constrains decoding to schema-consistent

grammar and alignment rules, reducing output uncertainty:

$$H(Y \mid X, z) \leq H(Y \mid X), \quad (3)$$

where $H(Y \mid X) - H(Y \mid X, z) = I(Y; Z \mid X)$ and $I(Y; Z \mid X)$ denotes the conditional mutual information measuring the additional information contributed by the structural schema beyond the visual input. This structural conditioning prunes invalid hypotheses and suppresses outputs that are locally plausible yet globally inconsistent. Overall, the multi-stage decomposition reformulates end-to-end transcription into semantically explicit, structure-controlled steps, improving interpretability while substantially enhancing decoding stability and robustness.

### 4.3. Schema-Anchored In-Context Learning (SIL)

To robustly instantiate the latent structural variable $Z$ at inference time, we inject a lightweight in-context structural prior without introducing extra fine-tuning, changing system instructions or decoding settings. Concretely, we prepend a small set of anchor pairs $\mathcal{D}_{\text{anchor}} = \{(X^{(i)}, z^{(i)})\}_{i=1}^{K}$ to the prompt (Fig. 3), where $K$ is the number of anchor pairs, which stabilizes schema selection by calibrating the model's perception of layout patterns. This can be viewed as a prompt-induced approximation to amortised inference:

$$P(z \mid X, \mathcal{D}_{\text{anchor}}) \approx P(z \mid X). \quad (4)$$

$\mathcal{D}_{\text{anchor}}$ acts as a prompt-induced structural prior that stabilizes schema inference, increasing the posterior mass on the correct schema. Importantly, these anchor tuples are not meant to teach symbol-level knowledge or arithmetic rules. Instead, they serve as structural anchors that bias the model toward forming correct high-level layout assumptions, thereby reducing variance in schema prediction

under visually ambiguous inputs. Meanwhile, the generation of structured LaTeX remains uniformly governed by schema-specific syntax and formatting rules predefined in the prompt, which cleanly decouples schema inference from schema-constrained decoding.

This anchoring mechanism mitigates cascading failures caused by early structural misclassification: once an incorrect schema is activated, subsequent symbol decoding is forced into incompatible alignment/grammar constraints, amplifying downstream errors.

Finally, MSR and SIL address two orthogonal uncertainties in our latent-variable structured inference framework. MSR reduces ambiguity in conditional generation $P(Y \mid X, z)$ via schema-consistent decoding, while SIL stabilizes $P(z \mid X)$ via a prompt-induced prior. When $P(z \mid X)$ is sharply peaked at $\hat{z}$, we approximate $P(Y \mid X)$ by the dominant component:

$$P(Y \mid X) \approx P(Y \mid X, \hat{z}), \quad \hat{z} = \arg\max_{z \in \mathcal{Z}} P(z \mid X). \quad (5)$$

This formulation shows that HMER is influenced by structural uncertainty: once the schema is correctly inferred, the remaining transcription becomes substantially easier and more stable.

## 5. Experiments

### 5.1. Experimental Settings

**Models Under Evaluation.** We evaluate four groups of models: general-purpose VLMs, OCR-tuned VLMs, HMER-specific VLMs, and Mathematical Expression Recognition APIs (MER-APIs). These four categories represent widely accessible large-model-based or API-based recognition systems that can be evaluated without additional training.

For HMER-specific VLMs, we include Uni-MuMER in the main comparison because reproducible public weights are available. Other HMER-specific VLMs, such as HiE-VL and VLPG, are not included in the main comparison due to the lack of publicly reproducible weights, code, or sufficient implementation details for fair evaluation. In addition, supervised HMER systems, such as CoMER, SAN, TAMER, and PosFormer, provide important references for HMER. However, evaluating them on HMER-Bench in a fully comparable manner would require constructing benchmark-specific train/validation/test splits and retraining the models on our data. This would shift the evaluation setting from out-of-domain structural generalization to supervised benchmark fitting, which is not the main focus of this work.

For general-purpose VLMs and OCR-tuned VLMs that support custom prompts, we specify the annotation guidelines in the system prompt to standardize symbol conventions and structural constraints. For OCR-tuned VLMs that do not support custom prompts and for MER-APIs, we impose no additional constraints on output format or reasoning procedure. All VLMs that support prompting are provided with the same formatting instructions in both the baseline and SASR settings. Therefore, SASR does not receive additional syntactic information beyond what is given to the corresponding baseline. For models that do not support custom prompts, we consider the results on VAE and SDE not directly comparable, and thus exclude these two categories when computing the average.

**Our Framework.** We apply SASR to enhance reasoning on the Qwen3-VL-8B-Instruct (Bai et al., 2025; 2023; Yang et al., 2025a) model. SASR introduces no additional training data and involves no parameter updates; instead, it refines the original VLM's reasoning process during the testing phase.

**Evaluation Metrics. Expression Recognition Rate (ExpRate)** is the most commonly used metric in the field of HMER. It is defined as the percentage of images where the predicted sequence perfectly matches the true sequence out of the total number of images. **Character Detection Matching (CDM)** (Wang et al., 2025) renders predictions and references into images and matches symbol categories and their 2D relations, producing a per-sample Score@CDM. We report ExpRate@CDM by counting a prediction as correct only when its Score@CDM reaches 1.0, i.e., the prediction is exactly equivalent to the reference under CDM. This follows the strict equivalence setting of CDM and avoids introducing an additional tunable threshold.

### 5.2. Benchmark Results

**Quantitative Results.** Table 2 reports ExpRate on HMER-Bench, where the best average performance is achieved by Qwen3-VL-8B-Instruct with our SASR reasoning enhancement (24.52%). Table 3 presents results under the ExpRate@CDM metric, with Qwen3-VL-235B-A22B-Instruct reaching the highest average (34.45%). Despite these gains, overall VLM performance on HMER remains far from satisfactory: accuracy drops markedly on cases requiring complex 2D structure recovery, row/column alignment, and semantic symbol disambiguation, compared with short single-line expressions. Notably, some general-purpose VLMs generalize better to out-of-distribution structured formats (e.g., VAE and SDE), whereas OCR-tuned VLMs and HMER-specific VLMs often fail on these types. Without requiring training or additional data, SASR delivers stable gains across all categories. Its performance on complex 2D structures approaches or even surpasses that of the 235B-parameter Qwen3-VL, indicating that this method effectively enhances the model's ability to model and infer expression structural consistency.

*Table 2.* **ExpRate(%) on HMER-Bench.** The compared baselines include GOT-OCR2.0 (Wei et al., 2024), dots.ocr (Li et al., 2025b), DeepSeek-OCR (Wei et al., 2025), Phi-4-multimodal-instruct (Abouelenin et al., 2025), Ovis2.5 (Lu et al., 2025; 2024), and Qwen3-VL-235B-Instruct (Bai et al., 2025), among others. "-" indicates that the model does not support recognition for this category (counted as zero when computing the average), or that the parameter count is not reported.

| Model | SSE | LSE | ACE | RCE | SAE | CHEM | SLD | MLD | SDE | VAE | SOE | MAD | CME | Avg. | Params |
|---|---|---|---|---|---|---|---|---|---|---|---|---|---|---|---|
| *MER-APIs* | | | | | | | | | | | | | | | |
| SimpleTex | 33.65 | 17.90 | 3.40 | 0.00 | 7.83 | 0.00 | 29.10 | 0.88 | 0.00 | 0.00 | 37.42 | 14.06 | 2.12 | 13.94 | - |
| *HMER-specific VLMs* | | | | | | | | | | | | | | | |
| Uni-MuMER | 39.00 | 26.80 | 12.74 | 0.00 | 16.52 | 0.00 | 35.11 | 0.00 | 0.00 | 0.00 | 0.00 | 0.00 | 5.20 | 11.48 | 3B |
| *OCR-tuned VLMs* | | | | | | | | | | | | | | | |
| Qwen-VL-OCR | 25.50 | 9.84 | 10.79 | 0.00 | 10.09 | 0.09 | 19.74 | 0.00 | 0.00 | 0.00 | 20.83 | 9.91 | 4.90 | 9.46 | - |
| GOT-OCR2.0 | 26.73 | 6.65 | 3.40 | 0.00 | 4.17 | 0.00 | 11.07 | 0.00 | - | - | 0.00 | 0.00 | 0.44 | 4.43 | 0.58B |
| PaddleOCR-VL | 34.01 | 18.28 | 7.82 | 0.00 | 10.17 | 0.00 | 33.65 | 0.10 | - | - | 21.71 | 26.60 | 5.34 | 13.22 | 0.9B |
| HunyuanOCR | 41.37 | 27.65 | 11.72 | 0.00 | 15.22 | 0.00 | 40.60 | 0.00 | 0.00 | 0.00 | 0.00 | 0.00 | 3.95 | 11.89 | 1B |
| dots.ocr | 17.18 | 7.03 | 1.53 | 0.00 | 2.70 | 0.00 | 6.18 | 0.10 | - | - | 0.00 | 3.96 | 0.00 | 3.22 | 1.7B |
| DeepSeek-OCR | 0.26 | 0.00 | 0.08 | 0.00 | 0.00 | 0.00 | 0.00 | 0.00 | 0.00 | 0.00 | 0.09 | 0.00 | 0.00 | 0.04 | 3B |
| MonkeyOCR | 41.98 | 27.09 | 12.15 | 0.00 | 14.43 | 0.00 | 38.20 | 0.00 | - | - | 0.00 | 0.00 | 0.07 | 11.27 | 3B |
| Chandra | 2.54 | 1.31 | 0.85 | 0.00 | 1.13 | 0.00 | 2.83 | 0.00 | - | - | 11.65 | 21.42 | 1.83 | 3.57 | 9B |
| *General-purpose VLMs* | | | | | | | | | | | | | | | |
| Doubao-Seed-1.6 | 16.30 | 4.12 | 4.16 | 0.00 | 6.26 | 0.28 | 4.55 | 0.29 | 78.99 | 31.87 | 18.80 | 26.04 | 3.95 | 13.06 | - |
| Phi-4-multimodal-instruct | 3.94 | 1.22 | 0.93 | 0.00 | 0.96 | 0.00 | 1.29 | 0.10 | 0.00 | 0.29 | 4.85 | 0.75 | 0.37 | 1.24 | 5.6B |
| Ovis2.5-9B | 13.15 | 2.72 | 2.21 | 0.00 | 4.70 | 3.36 | 4.98 | 0.10 | 13.68 | 18.71 | 22.42 | 15.28 | 2.34 | 7.98 | 9B |
| Nemotron-Nano-12B-v2-VL | 23.93 | 12.00 | 3.91 | 0.00 | 8.09 | 0.00 | 18.28 | 0.00 | 0.00 | 0.00 | 0.26 | 0.94 | 1.90 | 5.86 | 12B |
| Llama 4 Maverick | 6.84 | 2.06 | 1.61 | 0.00 | 4.43 | 0.09 | 3.95 | 0.10 | 45.93 | 26.32 | 16.06 | 15.38 | 2.86 | 8.54 | 17B |
| Qwen3-VL-235B-Instruct | 32.78 | 16.12 | 11.47 | 0.00 | 13.39 | 1.40 | 20.17 | 1.17 | 78.83 | 37.43 | 30.36 | 31.04 | 6.81 | 20.21 | 235B |
| Qwen3-VL-8B-Instruct | 21.21 | 6.37 | 6.12 | 0.00 | 6.61 | 0.75 | 6.78 | 0.39 | 17.43 | 27.97 | 27.63 | 25.09 | 3.59 | 11.63 | 8B |
| Qwen3-VL-8B-Instruct(ours) | 38.65 | 18.09 | 10.45 | 0.00 | 16.00 | 8.13 | 28.58 | 18.67 | 70.20 | 37.91 | 42.72 | 32.45 | 8.27 | 24.52 | 8B |

*Table 3.* **ExpRate@CDM(%) on HMER-Bench** for categories where CDM-based visual equivalence evaluation is applicable.

| Model | SSE | LSE | ACE | RCE | SAE | CHEM | SLD | MLD | SOE | MAD | CME | Avg. | Params |
|---|---|---|---|---|---|---|---|---|---|---|---|---|---|
| *MER-APIs* | | | | | | | | | | | | | |
| SimpleTex | 40.40 | 23.34 | 4.08 | 0.00 | 9.48 | 8.88 | 32.02 | 35.97 | 40.69 | 29.06 | 3.00 | 21.17 | - |
| *HMER-specific VLMs* | | | | | | | | | | | | | |
| Uni-MuMER | 44.70 | 30.65 | 15.21 | 0.00 | 17.57 | 12.90 | 37.77 | 13.49 | 0.01 | 0.00 | 6.37 | 17.04 | 3B |
| *OCR-tuned VLMs* | | | | | | | | | | | | | |
| Qwen-VL-OCR | 42.94 | 21.27 | 19.71 | 0.00 | 18.70 | 16.26 | 32.70 | 47.02 | 35.57 | 21.98 | 12.59 | 25.33 | - |
| GOT-OCR2.0 | 29.10 | 7.87 | 3.65 | 0.00 | 4.43 | 4.11 | 12.19 | 11.44 | 0.62 | 3.40 | 0.73 | 7.29 | 0.58B |
| PaddleOCR-VL | 38.91 | 22.68 | 9.01 | 0.00 | 11.83 | 14.21 | 37.85 | 12.12 | 30.89 | 43.96 | 8.64 | 21.72 | 0.9B |
| HunyuanOCR | 47.41 | 35.33 | 13.76 | 0.00 | 17.30 | 15.14 | 46.44 | 49.36 | 50.49 | 22.17 | 9.15 | 28.80 | 1B |
| dots.ocr | 24.19 | 13.59 | 2.29 | 0.00 | 9.48 | 0.00 | 12.45 | 0.68 | 0.00 | 7.55 | 0.00 | 6.65 | 1.7B |
| DeepSeek-OCR | 0.88 | 0.28 | 0.08 | 0.00 | 0.26 | 0.09 | 0.43 | 0.20 | 0.09 | 0.00 | 0.07 | 0.23 | 3B |
| MonkeyOCR | 48.29 | 34.49 | 13.34 | 0.00 | 14.96 | 0.00 | 41.97 | 41.94 | 0.00 | 18.68 | 0.07 | 19.92 | 9B |
| Chandra | 5.52 | 6.56 | 2.12 | 0.00 | 6.96 | 1.50 | 9.44 | 18.18 | 14.65 | 32.45 | 3.59 | 9.34 | 9B |
| *General-purpose VLMs* | | | | | | | | | | | | | |
| Doubao-Seed-1.6 | 40.14 | 23.81 | 13.00 | 0.00 | 25.48 | 19.72 | 33.99 | 42.23 | 25.60 | 40.47 | 10.98 | 25.82 | - |
| Phi-4-multimodal-instruct | 10.25 | 3.66 | 1.53 | 0.00 | 3.13 | 3.55 | 3.95 | 7.33 | 6.88 | 1.32 | 0.66 | 3.96 | 5.6B |
| Ovis2.5-9B | 32.52 | 13.50 | 5.27 | 0.00 | 17.04 | 17.66 | 18.97 | 9.97 | 27.98 | 24.81 | 5.42 | 16.33 | 9B |
| Nemotron-Nano-12B-v2-VL | 28.40 | 16.78 | 5.18 | 0.00 | 9.39 | 11.31 | 24.29 | 9.87 | 12.53 | 16.13 | 3.44 | 12.95 | 12B |
| Llama 4 Maverick | 17.00 | 9.37 | 4.93 | 0.00 | 13.48 | 19.91 | 18.37 | 7.62 | 17.65 | 22.36 | 6.88 | 13.00 | 17B |
| Qwen3-VL-235B-Instruct | 51.53 | 35.15 | 19.46 | 0.00 | 34.87 | 31.96 | 43.61 | 52.69 | 35.39 | 45.85 | 16.18 | 34.45 | 235B |
| Qwen3-VL-8B-Instruct | 44.08 | 24.65 | 13.25 | 0.00 | 25.74 | 23.83 | 33.99 | 28.54 | 31.86 | 37.55 | 10.83 | 25.84 | 8B |
| Qwen3-VL-8B-Instruct(ours) | 48.38 | 27.09 | 13.93 | 0.01 | 24.00 | 25.42 | 38.03 | 50.64 | 45.45 | 35.57 | 11.27 | 29.98 | 8B |

**Qualitative Results.** Fig. 4 contrasts baseline renderings with SASR outputs on Qwen3-VL-8B-Instruct, highlighting recurring structure-related errors and the corresponding corrections induced by schema-anchored decoding. We defer detailed interpretation to Sec. 5.3.

## 5.3. Multi-Dimensional Performance Analysis

**Symbol-Level Accuracy vs. Structural Consistency.** The evaluation results of CDM metrics across the four dimensions of HMER-Bench for selected models are presented in Table 4. Across models, Score@CDM is consistently higher than ExpRate@CDM, indicating that many predictions are locally close to the reference after rendering-based matching but still fail to meet the strict whole-expression equivalence criterion. When considered together with the category-wise degradation on alignment-sensitive layouts and the qualitative examples in Fig. 4, the gap suggests that a substantial portion of failures arise from predictions that preserve many local symbols but violate global consistency constraints, such as two-dimensional layout, row-column alignment, and structure-conditioned symbol roles. Tables 2 and 3 show that incorporating SASR improves both ExpRate and ExpRate@CDM, with particularly pronounced gains in structurally demanding categories such as SDE. The

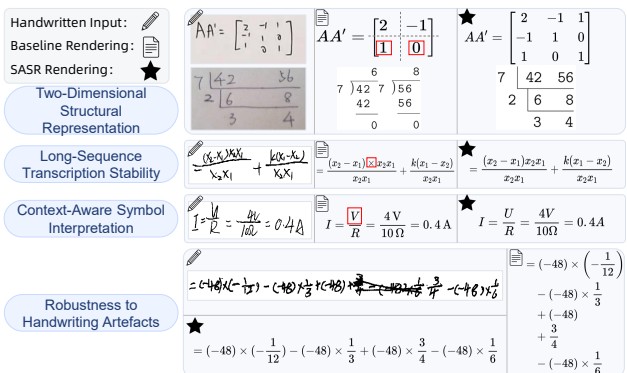

Handwritten Input:
Baseline Rendering:
SASR Rendering:

Two-Dimensional Structural Representation

Long-Sequence Transcription Stability

Context-Aware Symbol Interpretation

Robustness to Handwriting Artefacts

*Figure 4.* **Qualitative comparison on HMER-Bench.** We show representative baseline failures and the improvements from SASR.

*Table 4.* **Score@CDM (%) and ExpRate@CDM (%) across four dimensions for representative models on HMER-Bench (except SDE and VAE).** Score@CDM serves as an approximate proxy of character-level recognition accuracy under CDM-based visual equivalence. Qwen3-VL-8B denotes Qwen3-VL-8B-Instruct.

| Model metric | D1 Score | D1 Exp | D2 Score | D2 Exp | D3 Score | D3 Exp | D4 Score | D4 Exp | Avg. Score | Avg. Exp |
|---|---|---|---|---|---|---|---|---|---|---|
| SimpleTex | 89.57 | 32.16 | 91.60 | 34.47 | 81.84 | 9.19 | 75.46 | 2.83 | 85.47 | 21.17 |
| DeepSeek-OCR | 66.75 | 0.59 | 67.55 | 0.18 | 67.24 | 0.18 | 54.01 | 0.06 | 63.99 | 0.23 |
| Ovis2.5-9B | 83.59 | 23.32 | 79.26 | 20.61 | 80.42 | 17.34 | 65.37 | 3.65 | 77.38 | 16.33 |
| Qwen3-VL-8B | 87.76 | 34.69 | 87.01 | 33.03 | 86.90 | 24.82 | 72.13 | 9.19 | 84.17 | 25.84 |
| Qwen3-VL-8B(SASR) | 91.23 | 38.11 | 92.03 | 42.30 | 88.39 | 24.68 | 78.52 | 9.64 | 88.21 | 29.98 |

newly reported Score@CDM results in Table 4 further show that SASR does not eliminate the gap between local visual equivalence and strict expression-level correctness. Rather, its main effect is to convert a subset of locally plausible but globally inconsistent predictions into fully valid expressions through explicit schema inference and schema-constrained transcription. These results suggest that structural modeling and global consistency are important bottlenecks in complex handwritten expression recognition, in addition to local character recognition.

**Failure Mode Analysis.** *(1) Structural uncertainty is the primary source of transcription failures.* We analyze failure modes from both structural and semantic perspectives, focusing on how early structural uncertainty propagates through decoding and leads to transcription failures. Semantic symbol misinterpretation further exacerbates global inconsistency and degrades expression validity.

*(2) Models lack the ability to maintain stable global 2D structures in complex expressions.* Most models struggle to construct stable two-dimensional hierarchical representations for complex handwritten expressions. Common structural errors include layout category confusion (e.g., single-line vs. multi-line expressions, short vs. long division), incorrect row–column relations in matrices, and broken cross-line alignment in multi-line derivations. These issues intensify with increasing expression length and nesting depth, where attention often exhibits insufficient global coverage (symbol

*Table 5.* **Ablation of MSR and SIL on HMER-Bench (ExpRate, %)**: four diagnostic dimensions (D1–D4) and overall average.

| MSR | SIL | D1 | D2 | D3 | D4 | Avg. |
|---|---|---|---|---|---|---|
| ✗ | ✗ | 14.04 | 17.54 | 3.78 | 4.24 | 11.63 |
| ✓ | ✗ | 22.33 | 30.00 | 7.39 | 5.42 | $19.52^{+7.89}$ |
| ✗ | ✓ | 18.39 | 21.14 | 4.10 | 5.01 | $14.02^{+2.39}$ |
| ✓ | ✓ | 28.71 | 36.07 | 12.21 | 7.24 | $24.52^{+12.89}$ |

*Table 6.* **Ablation of MSR components on HMER-Bench (ExpRate, %).** Step1 denotes the structural schema inference and Step3 denotes the context-driven symbol disambiguation.

| Method | D3 ↑ | Avg. Exp ↑ |
|---|---|---|
| Baseline | 3.78 | 11.63 |
| MSR w/o Step1 | 4.23(+0.45) | 10.28(-1.35) |
| MSR w/o Step3 | 5.27(+1.49) | 18.84(+7.21) |
| MSR (full) | 7.39(+3.61) | 19.52(+7.89) |

or substructure omissions) and attention drift (repetitions or fragment reordering), revealing limited ability to maintain long-range structural coherence. Handwriting artefacts further aggravate this problem by introducing spurious marks or triggering incorrect schema activation.

*(3) Local symbol correctness does not guarantee global semantic validity.* When symbol interpretation depends jointly on global structure and contextual constraints, models frequently produce *locally plausible but globally invalid* predictions. For example, visually similar symbols are misclassified under discipline-specific contexts, and functional markings in reduction expressions (RCE) are confused with noise in artefact-corrupted samples (ACE), resulting in correct local shapes but incorrect global semantics. These patterns explain why models perform well on simple cases yet fail under complex structures: the dominant bottleneck lies in the absence of explicit global structural constraints and context-consistent inference.

*(4) Structural instability also triggers systematic instruction-following failures.* We further observe instruction-following failures under structural instability. Despite explicit prompts specifying output formats and annotation rules, models may generate explanatory text, meaningless tokens, or violate structured notation (e.g., vertical arithmetic or short division), which directly breaks downstream parsing and turns partially correct predictions into complete failures.

### 5.4. Ablation Study

**Component Contribution.** We ablate MSR and SIL in Table 5. Relative to the baseline (11.63%), MSR raises ExpRate to 19.52%, indicating that explicit structural conditioning curbs error propagation under complex 2D layouts; SIL alone reaches 14.02%, showing that a few structure anchors can calibrate structure–schema alignment without training data. The strongest improvements on D2 indicate

*Table 7.* **Performance–cost trade-off on HMER-Bench.** TotTok and Time are averaged per image. $\Delta$ values are computed w.r.t. Baseline. $\Delta$Exp/1kTok denotes ExpRate-point gain per additional 1k tokens, and $\Delta$Exp/sec per additional second.

| Method | Exp↑ | TotTok↓ | Time(s)↓ | ΔExp↑ | ΔTok↓ | ΔTime↓ | ΔExp/1kTok↑ | ΔExp/sec↑ |
|---|---|---|---|---|---|---|---|---|
| Baseline | 10.99 | 627.40 | 1.1044 | – | – | – | – | – |
| Simple CoT | 11.36 | 733.90 | 1.3034 | +0.37 | +106.50 | +0.1990 | 3.47 | 1.86 |
| Ours (SASR) | 23.24 | 3474.82 | 2.5570 | +12.25 | +2847.42 | +1.4526 | 4.30 | 8.43 |

*Table 8.* **Robustness to Gaussian noise (ExpRate, %).** $\sigma$: Clean$\in$ $\{0\}$; Mild$\in \{3, 5, 8\}$; Medium$\in \{10, 15, 20\}$; Severe$\in \{25, 30\}$. Rel.Drop $= (\max - \min)/\max$, where max / min are the best/-worst ExpRate across all $\sigma$ settings.

| Method | Clean | Mild Avg | Medium Avg | Severe Avg | Avg | Worst | Rel.Drop↓ |
|---|---|---|---|---|---|---|---|
| Baseline | 10.62 | 10.91 | 10.14 | 9.81 | 10.38 | 9.50 | 15.78% |
| SASR (Ours) | 23.53 | 23.55 | 23.21 | 21.86 | 23.06 | 21.60 | 9.36% |

*Table 9.* **Baseline vs. SASR ExpRate (%) on Ovis2.5-8B, Nemotron-Nano-12B-v2-VL, Kimi-K2.5 and GLM-4.6v**, evaluated on HMER-Bench dimensions D1–D4 and the overall average.

| Method | D1 | D2 | D3 | D4 | Avg. |
|---|---|---|---|---|---|
| Ovis2.5-8B(baseline) | 9.09 | 11.17 | 5.86 | 1.18 | 7.72 |
| Ovis2.5-8B(SASR) | **10.00** | **19.83** | **7.66** | **3.55** | **12.55** |
| Nemotron-Nano-12B-v2-VL(baseline) | 20.00 | 3.50 | 5.41 | 0.59 | 6.01 |
| Nemotron-Nano-12B-v2-VL(SASR) | **21.82** | **8.50** | **7.21** | **1.18** | **8.91** |
| Kimi-K2.5(baseline) | 34.55 | 40.17 | 10.36 | 10.65 | 28.06 |
| Kimi-K2.5(SASR) | **43.18** | **53.00** | **26.58** | **14.79** | **38.90** |
| GLM-4.6v(baseline) | 12.73 | 21.67 | 6.31 | 4.73 | 13.88 |
| GLM-4.6v(SASR) | **27.73** | **38.00** | **16.67** | **4.73** | **25.24** |

that our approach markedly enhances the model's reasoning capability for complex 2D layouts.

**Effect of Semantic Disambiguation.** We ablate the context-driven symbol disambiguation stage (Step3) in MSR under identical decoding settings. As shown in Table 6, Step1 (structural schema inference) is foundational: removing it reduces Avg. ExpRate from 11.63 to 10.28, suggesting that without a reliable schema anchor, constrained decoding can amplify misalignment. On top of Step1, Step3 yields complementary gains: MSR(full) improves D3 from 5.27 to 7.39 (+2.12, ∼40% relative) and also increases the overall Avg. ExpRate from 18.84 to 19.52 (+0.68). This indicates that Step3 resolves context-dependent symbol confusions, with benefits best realized under correct schema instantiation.

### 5.5. Efficiency, Robustness, and Generalization

**Time Efficiency.** To verify that the improvement does not stem from higher testing phase overhead, we compared metrics for Baseline, Simple CoT (simplifying input prompts while only outputting final results), and Ours (SASR) under identical transcription specifications. Table7 shows that Simple CoT increases token/latency by approximately 17%/18% while improving by only 0.37 percentage points, whereas SASR increases the ExpRate from 10.99% to 23.24% (+12.25) with higher efficiency, indicating that the gains primarily stem from our framework.

**Robustness to Gaussian Noise.** We add zero-mean Gaussian noise on the image (pixel values clipped to [0, 255] with multiple $\sigma$ and summarize ExpRate under mild/medium/severe regimes (Table 8). SASR maintains a consistently large margin across all noise levels and exhibits a smaller relative drop, whereas the baseline's modest absolute change is likely influenced by its low clean-set accuracy.

**SASR Generalization.** We evaluate the transferability of SASR by porting the same inference pipeline to Ovis2.5-8B, Nemotron-Nano-12B-v2-VL, Kimi-K2.5 (Team et al.,

2026), and GLM-4.6v (Hong et al., 2025). As shown in Table 9, SASR consistently improves Avg. ExpRate across all evaluated backbones: from 7.72 to 12.55 on Ovis2.5-8B (+4.83, +62.6%), from 6.01 to 8.91 on Nemotron-Nano-12B-v2-VL (+2.90, +48.3%), from 28.06 to 38.90 on Kimi-K2.5 (+10.84, +38.6%), and from 13.88 to 25.24 on GLM-4.6v (+11.36, +81.8%). The improvements are particularly pronounced on D2, which involves complex 2D structural representation, with gains of +8.66, +5.00, +12.83, and +16.33 on Ovis2.5-8B, Nemotron-Nano-12B-v2-VL, Kimi-K2.5, and GLM-4.6v, respectively. These results support that SASR is transferable across model families, although the gain magnitude varies with the base model's visual, structural, and instruction-following capabilities.

## 6. Conclusion

We present HMER-Bench, a structure-centric benchmark for evaluating structural reasoning in HMER. Our results show that structural inference, rather than symbol recognition, is the primary bottleneck of models. We further propose SASR, a training-free schema-anchored framework that enforces structure-consistent decoding and improves recognition accuracy and stability. We hope this work will facilitate future progress in HMER.

## Acknowledgements

This work is partially supported by grants from the National Natural Science Foundation of China under contract No. 62437001 and 62402051, and by the Fundamental Research Funds for the Central Universities.

## Impact Statement

This paper presents work whose goal is to advance the field of Machine Learning. There are many potential societal consequences of our work, none which we feel must be specifically highlighted here.

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

# A. Theoretical Justification of Structure-Aware Inference

### A.1. Notation

Let $X$ denote the input handwritten expression image, $Y$ the target LaTeX sequence, and $Z \in \mathcal{Z}$ a discrete latent variable representing the *structural schema* (e.g., matrix, fraction, multi-line derivation, vertical arithmetic). We assume a well-defined conditional model family $P(Y \mid X, z)$ and a schema posterior $P(z \mid X)$.

In MSR, intermediate variables $Y_2, Y_3, Y_4$ denote (i) an initial schema-conditioned transcription, (ii) a disambiguation signal, and (iii) the final structured output, respectively.

### A.2. Proposition 1 (Latent-variable decomposition)

For any discrete $Z \in \mathcal{Z}$,

$$P(Y \mid X) = \sum_{z \in \mathcal{Z}} P(Y \mid X, z)\, P(z \mid X). \tag{6}$$

**Proof.** By the law of total probability,

$$P(Y \mid X) = \sum_{z \in \mathcal{Z}} P(Y, z \mid X) = \sum_{z \in \mathcal{Z}} P(Y \mid X, z) P(z \mid X). \tag{7}$$

$\square$

### A.3. Proposition 2 (MSR chain-rule factorization)

For any random variables $(Z, Y_2, Y_3, Y_4)$,

$$P(Y_4, Y_3, Y_2, z \mid X) = P(z \mid X)\, P(Y_2 \mid X, z)\, P(Y_3 \mid X, z, Y_2)\, P(Y_4 \mid X, z, Y_2, Y_3). \tag{8}$$

**Proof.** This is a direct application of the chain rule of conditional probabilities:

$$P(Y_4, Y_3, Y_2, z \mid X) = P(z \mid X)\, P(Y_2 \mid X, z)\, P(Y_3 \mid X, z, Y_2)\, P(Y_4 \mid X, z, Y_2, Y_3). \tag{9}$$

No conditional independence assumption is required. $\square$

### A.4. Theorem 1 (Structural conditioning reduces uncertainty)

For any $(X, Y, Z)$,

$$H(Y \mid X, Z) \leq H(Y \mid X), \tag{10}$$

and the gap equals the conditional mutual information:

$$H(Y \mid X) - H(Y \mid X, Z) = I(Y; Z \mid X) \geq 0. \tag{11}$$

**Proof.** By definition of conditional mutual information,

$$I(Y; Z \mid X) = H(Y \mid X) - H(Y \mid X, Z). \tag{12}$$

Since mutual information is non-negative, $I(Y; Z \mid X) \geq 0$, we obtain $H(Y \mid X, Z) \leq H(Y \mid X)$. $\square$

**Discussion.** In HMER, $Z$ captures 2D layout constraints; when $Z$ remains informative about $Y$ even after observing $X$ (i.e., $I(Y; Z \mid X)$ is large), conditioning on $Z$ yields a substantial entropy reduction, meaning fewer plausible hypotheses remain for decoding.

### A.5. Theorem 2 (MAP-schema approximation error bound)

Let $\hat{z} = \arg\max_{z \in \mathcal{Z}} P(z \mid X)$ and $\alpha := P(\hat{z} \mid X)$. Then for any output $Y$,

$$0 \leq P(Y \mid X) - \alpha P(Y \mid X, \hat{z}) \leq 1 - \alpha. \tag{13}$$

In particular, if $P(z \mid X)$ is sharply peaked so that $\alpha \approx 1$, the mixture $P(Y \mid X) = \sum_z P(Y \mid X, z) P(z \mid X)$ is dominated by the MAP component.

**Proof.** Start from the mixture:

$$P(Y \mid X) = \sum_{z \in \mathcal{Z}} P(Y \mid X, z) P(z \mid X) \tag{14}$$

$$= \alpha P(Y \mid X, \hat{z}) + \sum_{z \neq \hat{z}} P(Y \mid X, z) P(z \mid X). \tag{15}$$

The remainder term is nonnegative, hence $P(Y \mid X) \geq \alpha P(Y \mid X, \hat{z})$, giving the left inequality. Moreover, because $P(Y \mid X, z) \leq 1$ for any $z$,

$$\sum_{z \neq \hat{z}} P(Y \mid X, z) P(z \mid X) \leq \sum_{z \neq \hat{z}} P(z \mid X) = 1 - \alpha, \tag{16}$$

which yields the right inequality. $\square$

### A.6. Theorem 3 (Error decomposition: schema vs. transcription)

Let $z^*(X)$ denote the (task-defined) correct schema for input $X$, $\hat{z}(X)$ the inferred schema, and $\hat{Y}(X)$ the final output produced by MSR. Then the overall failure probability admits the union-bound decomposition:

$$\Pr\left(\hat{Y}(X) \neq Y^*(X) \mid X\right) \leq \Pr(\hat{z}(X) \neq z^*(X) \mid X) + \Pr\left(\hat{Y}(X) \neq Y^*(X) \mid X, \hat{z}(X) = z^*(X)\right). \tag{17}$$

where the first term corresponds to schema inference error (targeted by SIL), and the second corresponds to schema-conditioned transcription/disambiguation error (targeted by MSR).

**Proof.** Define the failure event $F = \{\hat{Y} \neq Y^*\}$. We have the set inclusion

$$F \subseteq \{\hat{z} \neq z^*\} \cup \{\hat{Y} \neq Y^*, \hat{z} = z^*\}. \tag{18}$$

Taking probabilities conditioned on $X$ and applying the union bound,

$$\Pr(F \mid X) \leq \Pr(\hat{z} \neq z^* \mid X) + \Pr(\hat{Y} \neq Y^*, \hat{z} = z^* \mid X) \tag{19}$$

$$\leq \Pr(\hat{z} \neq z^* \mid X) + \Pr(\hat{Y} \neq Y^* \mid X, \hat{z} = z^*). \tag{20}$$

$\square$

### A.7. Remark on SIL as a prior calibration mechanism

SIL prepends anchor pairs $\mathcal{D}_{anchor}$ to stabilize schema inference. In general, conditioning on $\mathcal{D}_{anchor}$ changes the induced posterior, so $P(z \mid X, \mathcal{D}_{anchor}) \neq P(z \mid X)$ in principle. A precise statement is that anchors act as a prompt-induced prior that aims to reduce schema inference error $\Pr(\hat{z} \neq z^* \mid X)$ (Theorem A.6), and to increase the MAP mass $\alpha = P(\hat{z} \mid X, \mathcal{D}_{anchor})$ (Theorem A.5), thereby tightening the MAP approximation.

## B. HMER-Bench Details

### B.1. Category Design and Capability Coverage

HMER-Bench encompasses mathematical expressions commonly encountered across primary, secondary, and tertiary education. Sample images and their corresponding ground truth for each category are illustrated in Fig. 5, with the core challenges outlined as follows.

**Short Single-Line Expressions** and **Long Single-Line Expressions.** Sequence length serves as a core metric for evaluating the effectiveness of Transformer architecture attention mechanisms. As sequence length increases, attention weights become dispersed, leading to word omission or repetition phenomena. By strictly partitioning expressions into short-sequence and long-sequence groups, we can quantify the model's performance degradation over long-range dependencies.

**Artifact Corrupted Expressions.** This category covers varying degrees of student edits and artifacts, including strike-through deletions, blacked-out regions, and underlines. It evaluates whether a model can ignore these irrelevant marks and

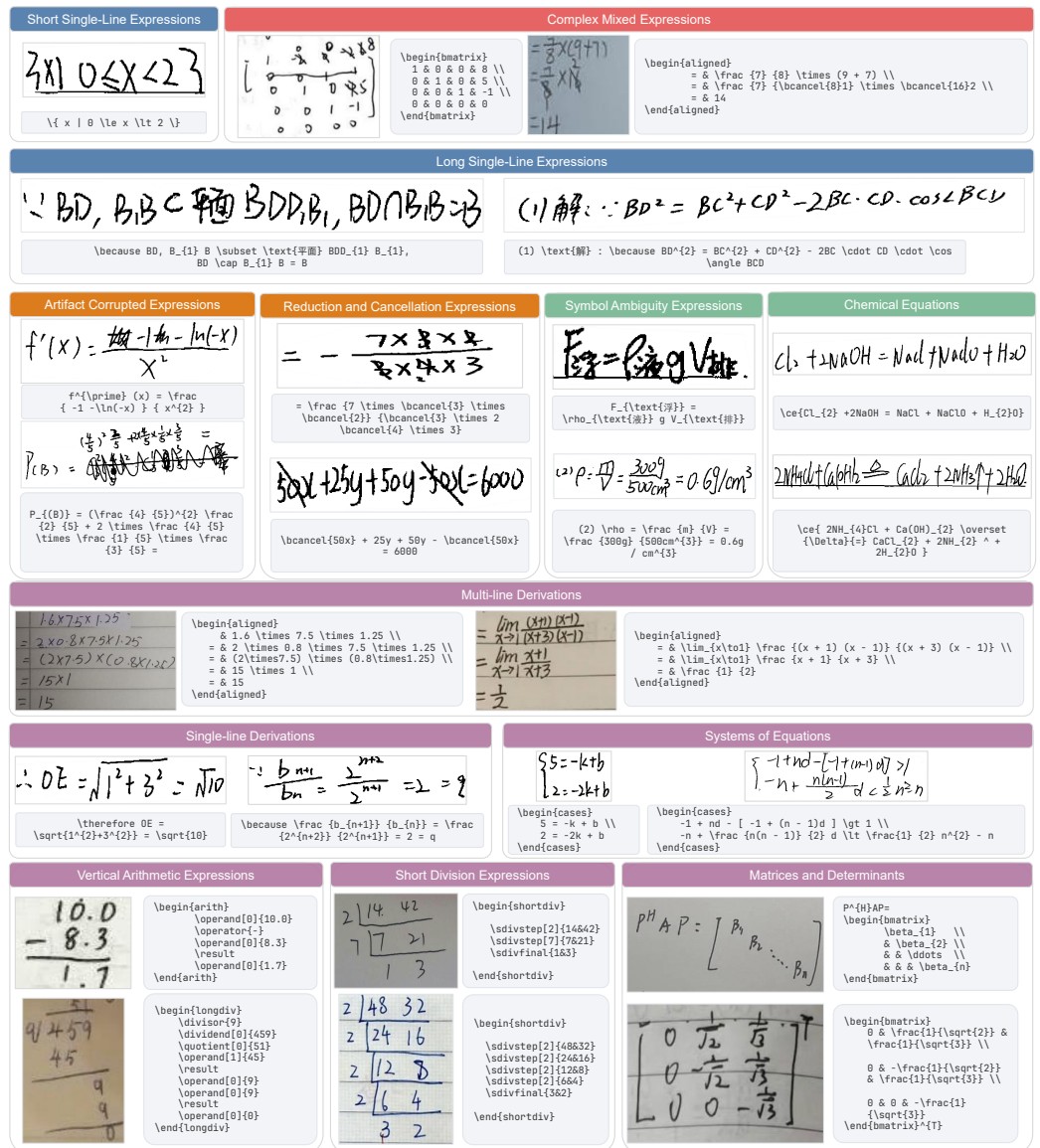

*Figure 5.* Representative examples from all HMER-Bench categories with their corresponding ground-truth annotations.

focus solely on the valid handwriting. For traditional OCR systems that rely on connected-component analysis, such artifacts can severely disrupt segmentation and region grouping; for end-to-end models, this setting primarily tests the encoder's ability to extract clean and stable representations under clutter, i.e., to remain robust to edit-induced noise without being distracted by it.

**Reduction and Cancellation Expressions.** Unlike the strikethroughs denoting smudging or deletion in Artifact Corrupted Expressions, the strikethroughs within Reduction and Cancellation Expressions serve to express mutual cancellation or simplification. They typically appear in paired form and maintain a close semantic association with the affected symbols. Consequently, the model must explicitly output them as structural information. The model must comprehend this writing operation: certain strokes constitute not the characters themselves to be recognised, but rather symbolic markers used to modify, negate, or erase other strokes. This challenge transcends mere character recognition, demanding instead the inference and reconstruction of the writer's intent.

**Symbol Ambiguity Expressions.** In mathematical expressions, many symbols are visually confusable, while their semantics are strongly constrained by context. For instance, handwritten $w$, $\omega$, $p$, and $\rho$ can be difficult to distinguish purely from

local appearance, yet the surrounding physical/mathematical context often admits a more plausible interpretation: in buoyancy-related expressions, the symbol is more likely to be $\rho$ (density), whereas in pressure expressions it is more likely to be $p$. This category is designed to evaluate whether models can leverage contextual cues for symbol disambiguation, rather than relying solely on local visual features. As a result, the HMER task is pushed beyond pure visual transcription toward context-aware decision making and semantic-consistency modeling.

**Chemical Equations.** This category is similar to Symbol Ambiguity Expressions in that it evaluates whether models can leverage context for symbol disambiguation. The key difference is that the contextual dependency here is shorter, with disambiguation primarily driven by local symbol co-occurrence patterns and compositions.

**Single-line Derivations** and **Multi-line Derivations.** Single-line Derivations evaluate a model's ability to segment derivation chains and model logical relations along the horizontal direction. In contrast, Multi-line Derivations assess vertical reading-order inference, cross-line alignment, and inter-line dependency modeling. These capabilities are essential for accurately parsing long-form mathematical derivations and proofs.

**Short Division Expressions.** SDE are among the most common and fundamental handwritten arithmetic formats in basic education. They feature a characteristic step-wise topology, strong semantic dependencies across lines, and strict column-alignment constraints. Accordingly, this category primarily evaluates a model's ability to jointly model 2D structure, alignment relations, and cross-line dependencies.

**Vertical Arithmetic Expressions.** VAE are a core calculation format in basic education. They are characterized by a quasi-grid layout and cross-line right alignment with indentation. This category primarily evaluates a model's ability to model such 2D structure and alignment constraints.

**Systems of Equations.** SOE typically consist of a left curly brace and multiple equation lines, with explicit alignment and correspondence across lines. This category primarily evaluates a model's ability to model multi-line block structures and to understand and reconstruct cross-line alignment constraints.

**Matrices and Determinants.** MAD typically consist of paired delimiters (e.g., parentheses or brackets) enclosing an implicit grid of entries. This category primarily evaluates a model's ability to model the overall structure and to recover row–column grid alignment and correspondence.

**Complex Mixed Expressions.** CME integrate and nest multiple expression types from the above categories, often combining diverse structural constraints and semantic dependencies. This category evaluates a model's overall parsing capability as well as its robustness under complex mixed scenarios.

## B.2. Annotation Process and Quality Control

HMER-Bench was annotated with the support of a professional annotation company. The full process covered about 140,000 samples, from which we selected 13,513 representative images for the final benchmark. Reliability was ensured through standardized guidelines, company-side quality control, and two rounds of full manual verification by our team, with mandatory re-annotation for non-compliant samples. Under a strict criterion where even a single-character error counts as incorrect, the pass rate reached 99% after the second round. We also normalized labels (e.g., \ne/\neq, exponent notation) to reduce non-essential inconsistencies.

## B.3. Length Threshold Selection via Token-Length Profiling

To make the length split between SSE and LSE in our benchmark more well-grounded, we evaluate performance on HME100K by grouping samples according to LaTeX token length.

The Fig. 6 reports results on HME100K across three representative model families: traditional HMER models, OCR-tuned VLMs, and general-purpose VLMs. We observe a pronounced drop in recognition accuracy when the sequence length falls in the 30–35 token range, where the ExpRate consistently remains below 60% for all three model types. Therefore, we adopt 30 tokens as a reasonable threshold for separating short and long single-line expressions.

## B.4. Category Statistics and Data Sources of HMER-Bench

The per-category sample counts and data sources of HMER-Bench are summarized in Table 10. The benchmark comprises 13 categories with 13,513 samples in total: most categories are collected from middle-school exam answer sheets (covering

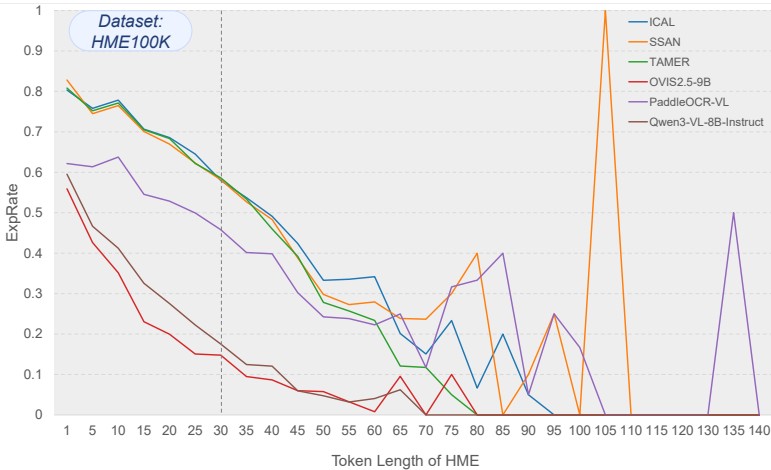

*Figure 6.* ExpRate on HME100K as a function of LATEX token length (binned), comparing traditional HMER models, OCR-tuned VLMs, and general-purpose VLMs.

short/long single-line expressions, artifact-corrupted cases, reduction/cancellation, symbol ambiguity, chemical equations, single-line derivations, and systems of equations), reflecting realistic educational scenarios. We further incorporate cropped and re-annotated $M^2E$ samples to enrich multi-line derivations and vertical arithmetic structures, and complement the benchmark with web-sourced short division cases, matrix analysis homework for matrices and determinants, and mixed sources for complex mixed expressions, improving the diversity of layouts and writing conditions.

*Table 10.* Category statistics of HMER-Bench: sample count, and data source.

| Category | Count | Source |
|---|---|---|
| Short Single-Line Expressions | 1141 | Middle school exam answer sheets |
| Long Single-Line Expressions | 1067 | Middle school exam answer sheets |
| Artifact Corrupted Expressions | 1177 | Middle school exam answer sheets |
| Reduction and Cancellation Expressions | 521 | Middle school exam answer sheets |
| Symbol Ambiguity Expressions | 1150 | Middle school exam answer sheets |
| Chemical Equations | 1070 | Middle school exam answer sheets |
| Single-line Derivations | 1165 | Middle school exam answer sheets |
| Multi-line Derivations | 1023 | Web/$M^2E$(cropped & re-annotated) |
| Short Division Expressions | 614 | Web |
| Vertical Arithmetic Expressions | 1026 | $M^2E$(cropped& re-annotated) |
| Systems of Equations | 1133 | Middle school exam answer sheets |
| Matrices and Determinants | 1060 | Matrix Analysis homework |
| Complex Mixed Expressions | 1366 | Mixed sources |

## C. Comparison with Existing HMER Datasets

Figure 7 compares HMER-Bench with representative HMER datasets, including CROHME, HME100K, and $M^2E$. Our claim is not that structural inference is the primary bottleneck on all HMER datasets, but that it becomes a major bottleneck under the real educational distribution covered by HMER-Bench.

Existing datasets emphasize different aspects of HMER. CROHME mainly focuses on relatively compact single-line expressions, where the structural search space is limited. HME100K contains real handwritten expressions, but provides limited coverage of alignment-intensive educational layouts. $M^2E$ supports two-dimensional handwritten math and is useful for multi-line content. However, as shown in Figure 8, its annotation target is closer to line-wise or segmented recognition: structure type, row–column alignment, and cross-line relations are not explicitly encoded in a unified whole-expression representation.

In contrast, HMER-Bench is designed for overall 2D structure recovery. It uses unified, renderable LaTeX annotations to preserve full-structure semantics and alignment constraints in categories such as multi-line derivations, vertical arithmetic,

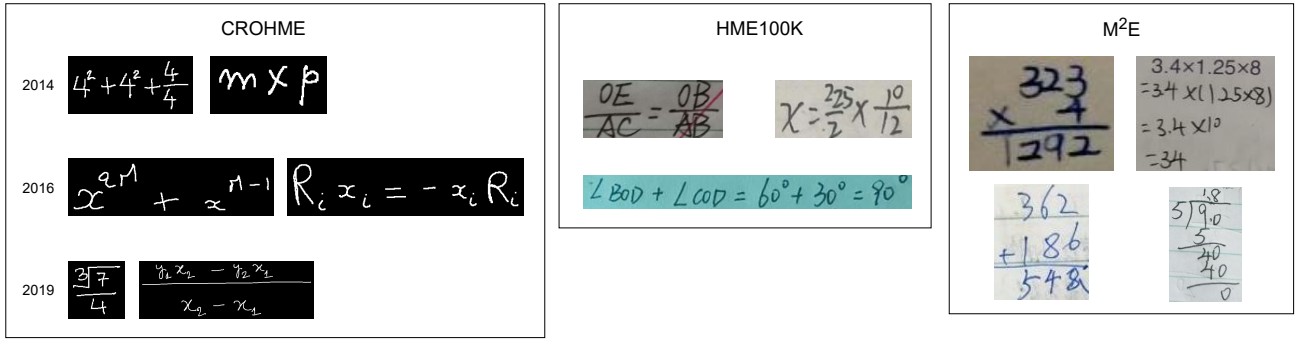

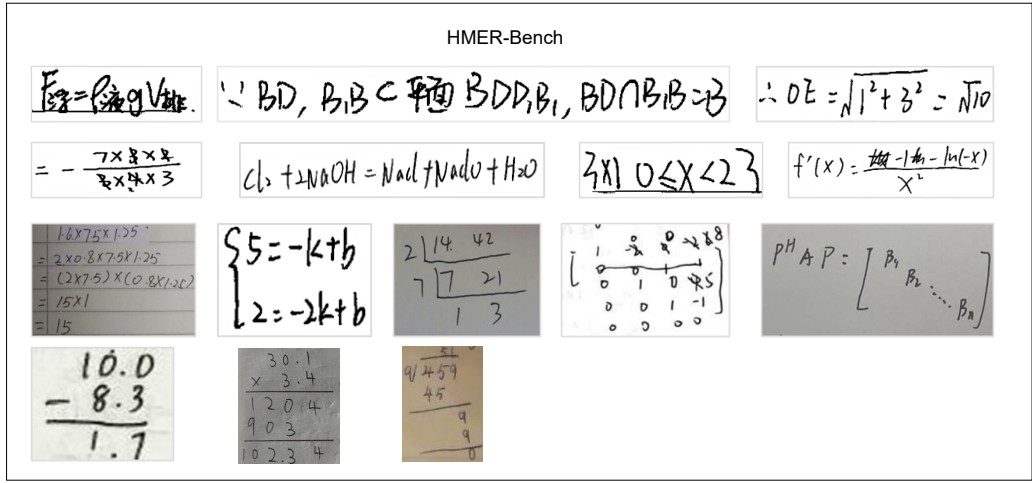

*Figure 7.* Visual comparison between HMER-Bench and representative HMER datasets.

short division, matrices and determinants, and systems of equations. Therefore, HMER-Bench complements prior datasets by focusing on structure-intensive educational cases where global 2D layout, cross-line correspondence, and row–column alignment are central to evaluation.

This distinction also explains the scope of SASR. Since SASR first infers a structural schema and then constrains transcription accordingly, it mainly benefits settings where global 2D structure is present and evaluation-relevant, rather than all HMER settings equally.

# D. arithstudio.sty: Formal Specification and Usage

### D.1. Motivation and Design Principles

Existing HMER benchmarks typically use LaTeX as the ground-truth label. However, for educational arithmetic layouts such as vertical calculations and (short/long) division, standard LaTeX math environments provide no native, canonical representation. As a result, these non-standard 2D layouts (Fig. 9) are difficult to encode in a standardized and faithfully renderable form using existing commands. To address this issue, we propose a unified annotation standard tailored to arithmetic layout structures, implemented as a renderable LaTeX macro package, `arithstudio.sty`. This annotation standard specifically describes spatial organisational relationships within vertical calculations and short division, enabling their accurate representation, stable rendering, and utilisation for model evaluation.

This specification follows three design principles: (1) Structural explicitisation: by explicitly declaring the spatial relationships between operands, operators, and intermediate results, it preserves row/column alignment information rather than compressing it into a linear sequence; (2) Semantic consistency: the annotation aims not merely at character recognition, but also at preserving the correctness of the operation order and the resulting mathematical semantics; (3) Implementation feasibility: realised with standard LaTeX mechanisms, it avoids reliance on external rendering engines, ensuring the annotations can be directly used for data release and evaluation.

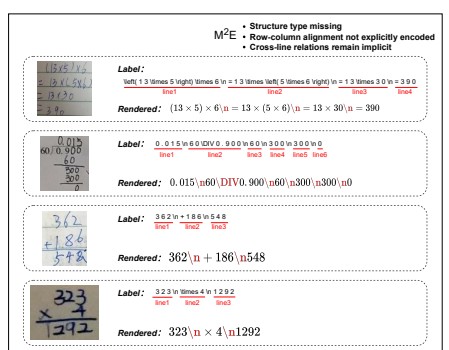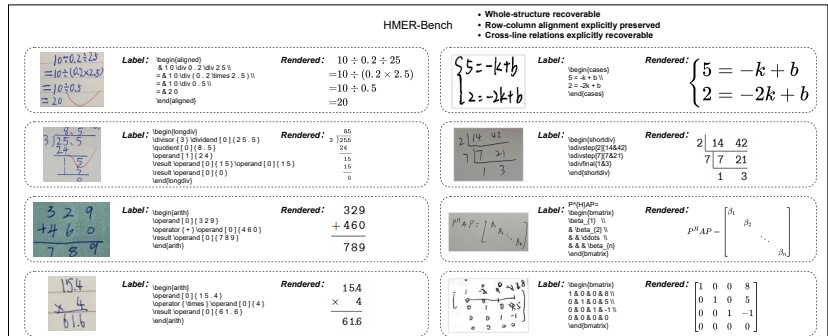

*Figure 8.* Annotation comparison between M$^2$E and HMER-Bench.

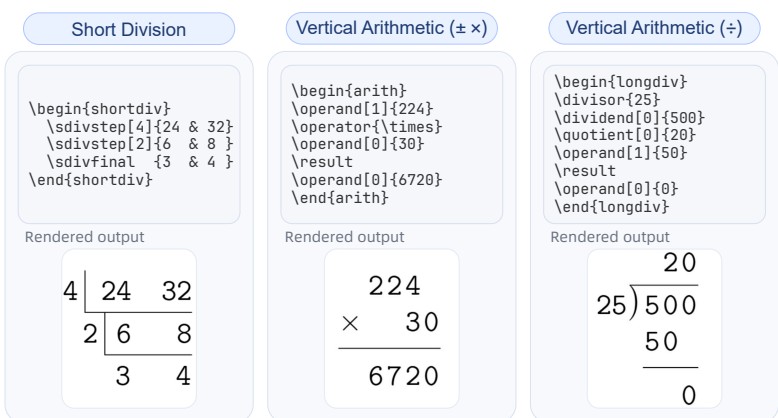

*Figure 9.* Annotation examples and rendering results adhering to the arithstudio.sty specification

## D.2. Syntax and Semantics: `arith`, `longdiv`, and `shortdiv`

We define three environments, `arith`, `longdiv`, and `shortdiv`. Here, $X^*$ denotes zero or more repetitions of $X$, and $X^+$ denotes one or more repetitions.

**Vertical arithmetic (addition/subtraction/multiplication)** For arithmetic operations involving addition, subtraction, and multiplication, we define an `arith` environment that converts the visual 2D layout into a 1D semantic tree. Formally,

$$\mathcal{A} ::= \texttt{arith}\langle S^+ \rangle, \tag{21}$$

$$S ::= \texttt{operand}[i]\{n\} \mid \texttt{operator}\{o\} \mid \texttt{result}. \tag{22}$$

Here, `\operand[i]{n}` declares an operand (or an intermediate value). Since vertical arithmetic is typically right-aligned, the optional parameter `[i]` specifies the character-level indentation measured from right to left. The `arith` environment parses all `\operand` tokens to infer the layout width automatically.

`\operator{o}` declares an operator (e.g., +, -, ×). It is automatically positioned to the left of the subsequent `\operand` and vertically aligned with it.

`\result` is a declarative marker with no arguments: it draws a horizontal rule above the next `\operand`, explicitly marking the next `\operand` as the computed result. The rule length is determined automatically based on the maximum width among all operands.

**Long division** For long division, we define:

$$\mathcal{L} ::= \mathtt{longdiv}\langle\mathtt{divisor}\{d\}\ \mathtt{dividend}[i]\{n\}\ \mathtt{quotient}[i]\{q\}\ S^*\rangle, \tag{23}$$

where \divisor{d} denotes the divisor positioned on the left, semantically serving as the root node. \dividend[i]{n} represents the dividend, typically marked with an overline. \quotient[i]{q} denotes the quotient, positioned above the dividend and aligned with it via the [i] parameter. $S^*$ represents recursively containing \operand and \result, describing the multiplication and subtraction operations at each step. We reuse the same token set $S$ to describe intermediate operands and result separators within longdiv.

**Short division** Short division features a distinctive "L"-shaped bracket structure. We define it as:

$$\mathcal{S} ::= \mathtt{shortdiv}\langle(\mathtt{sdivstep}[d]\{c_1\&c_2\&\ldots\})^+\ \mathtt{sdivfinal}\{r_1\&r_2\&\ldots\}\rangle. \tag{24}$$

Here, & separates columns. \sdivstep[d] specifies the divisor d for each layer, while \sdivfinal denotes the final row of coprime results.

Overall, this annotation standard reduces visual ambiguity while remaining fully renderable: by importing arithstudio.sty, the annotations can be compiled back into high-quality images. This also enables scalable synthesis for future large-scale pre-training.

## E. Novelty and Discussion of SASR

**Novelty.** We recast HMER as structure-conditioned transcription by introducing a latent structural schema $Z$ and decomposing $P(Y \mid X)$ into schema inference $P(z \mid X)$ and schema-conditioned generation $P(Y \mid X, z)$. This makes global 2D layout a first-class variable rather than an implicit byproduct of left-to-right decoding, which prior works note is prone to structural inconsistency under complex layouts.

**MSR (inference-time structural gate).** Unlike training-centric structure injection (e.g., grammar/tree-aware decoding or auxiliary tree scoring), MSR enforces "schema-first" decoding: the predicted $z$ directly restricts downstream grammar/alignment, pruning locally plausible yet globally invalid hypotheses and reducing uncertainty in $P(Y \mid X, z)$

**SIL (prompt-induced structural prior).** SIL stabilizes $P(z \mid X)$ via a few $(X^{(i)}, z^{(i)})$ anchors, providing a lightweight structural prior without fine-tuning—distinct from approaches that rely on architecture/training to encode structural representations (e.g., PosFormer's explicit structural encoding and joint optimization).

**Why MSR+SIL.** The two modules reduce orthogonal uncertainties: SIL improves schema selection $P(z \mid X)$, while MSR improves conditional decoding $P(Y \mid X, z)$, mitigating cascading failures caused by early structural misclassification.

## F. Prompt Design

For the general-purpose VLM and the OCR-tuned VLM supporting custom prompts, we specify the benchmark annotation standards within the system prompt. This includes the LaTeX representation standards for vertical arithmetic and short division proposed herein, ensuring the models could perform HMER tasks under explicit symbolic and structural conventions.

---

**Prompt with Annotation Guidelines**

**System Prompt:**

```
You are a LaTeX transcription expert.
CRITICAL: You must use the following CUSTOM SYNTAX for specific arithmetic structures:
1. Vertical arithmetic (+,-,\times): \begin{arith}...\end{arith}, each command auto-breaks line. \
operand[indent]{num} ([indent]=right-align level, 0=none), \operator{symbol} (+,-,\times), \
result (horizontal line). Ex: \begin{arith}\operand[0]{123}\operator{+}\operand[0]{456}\result\
operand[0]{579}\end{arith}
2. Long division: \begin{longdiv}...\end{longdiv}, each command auto-breaks line. \divisor{num}, \
dividend[indent]{num} (top), \quotient[indent]{num} (result above), \operand[indent]{num} (steps),
 \result (line). Ex: \begin{longdiv}\divisor{15}\dividend[1]{138}\quotient[0]{9.2}\operand
[1]{135}\result\operand[0]{30}\operand[0]{30}\result\operand[0]{0}\end{longdiv}
```

---

```
3. Short division: \begin{shortdiv}...\end{shortdiv}, each command auto-breaks line. \sdivstep[
divisor]{n1&n2&...} ([divisor]=divisor num, & aligns columns), \sdivfinal{n1&n2&...} (final row).
 Ex: \begin{shortdiv}\sdivstep[2]{12&18}\sdivstep[3]{6&9}\sdivfinal{2&3}\end{shortdiv}
4. Matrices: \begin{bmatrix}/\begin{pmatrix}/\begin{vmatrix}
5. Equation systems: \begin{cases}
6. Multi-line equations: \begin{aligned}
7. Chemical equations: \ce{...}
```

**User Prompt:**

```
Please transcribe all content in this image using the custom LaTeX standards defined above.
Output strictly the result enclosed in \boxed{}. Do not output any markdown code blocks (```latex)
.
```

**Answer:**

```
\boxed{B = (-\infty, 0) \cup (2, +\infty)}
```

For the OCR-tuned VLM without custom prompt support and the MER-API, no additional constraints were imposed on their output formats or inference processes.

## Prompt Without Additional Constraints

**System Prompt:**

```
You are a LaTeX transcription expert.
```

**User Prompt:**

```
I have an image of a handwritten mathematical expression. Please write out the expression of the
formula in the image using LATEX format
```

**Answer:**

```
B = (-\infty, 0) \cup (2, +\infty)
```

Our method, Schema-Anchored Structure-Aware Reasoning, builds upon the prompt with annotation guidelines by incorporating a series of prompts to support structure-aware and semantic dependency parsing.

## Prompt Without Additional Constraints

**System Prompt:**

```
You are a LaTeX transcription expert.
CRITICAL: You must use the following CUSTOM SYNTAX for specific arithmetic structures:
1. Vertical arithmetic (+,-,\times): \begin{arith}...\end{arith}, each command auto-breaks line. \
operand[indent]{num} ([indent]=right-align level, 0=none), \operator{symbol} (+,-,\times), \
result (horizontal line). Ex: \begin{arith}\operand[0]{123}\operator{+}\operand[0]{456}\result\
operand[0]{579}\end{arith}
2. Long division: \begin{longdiv}...\end{longdiv}, each command auto-breaks line. \divisor{num}, \
dividend[indent]{num} (top), \quotient[indent]{num} (result above), \operand[indent]{num} (steps),
 \result (line). Ex: \begin{longdiv}\divisor{15}\dividend[1]{138}\quotient[0]{9.2}\operand
[1]{135}\result\operand[0]{30}\operand[0]{30}\result\operand[0]{0}\end{longdiv}
3. Short division: \begin{shortdiv}...\end{shortdiv}, each command auto-breaks line. \sdivstep[
divisor]{n1&n2&...} ([divisor]=divisor num, & aligns columns), \sdivfinal{n1&n2&...} (final row).
 Ex: \begin{shortdiv}\sdivstep[2]{12&18}\sdivstep[3]{6&9}\sdivfinal{2&3}\end{shortdiv}
4. Matrices: \begin{bmatrix}/\begin{pmatrix}/\begin{vmatrix}
5. Equation systems: \begin{cases}
6. Multi-line equations: \begin{aligned}
7. Chemical equations: \ce{...}
```

**User Prompt:**

```
    Please transcribe this handwritten mathematical image using the following Chain of Thought
    process:

    Step 1: Formula Type Classification
    Carefully analyze the image and classify the formula type into ONE of these categories:
    1. Long Division
    2. Short Division
    3. Vertical Arithmetic
    4. Matrix
    5. Equation System
    6. Multi-line Derivations
    7. Single-line Derivations
    8. Chemical Equation
    9. Cancellation/Simplification
    10. Artifact Corrupted formula
    11. Single-line formula

    Output your classification as: <CLASSIFICATION>category_name</CLASSIFICATION>

    Step 2: Initial Transcription
    Transcribe the formula using appropriate LaTeX syntax based on your classification.
    Output as: your_latex_code

    Step 3: Semantic Disambiguation
    Use context to disambiguate symbols. For example:
    In velocity formulas, 'v' is usually velocity, not 'V'.
    In circular formulas, '\omega' is angular velocity, not 'w'.
    Output result as: <DISAMBIGUATION>explanation</DISAMBIGUATION>

    Step 4: Final Output
    Based on the disambiguation, output the final corrected LaTeX enclosed in \boxed{}.
    Do NOT output markdown code blocks (no ```latex).
    Output only: \boxed{your_final_latex_here}
```

**Answer:**

```
    <CLASSIFICATION>Single-line formula</CLASSIFICATION>\n$B=(-\infty,0)\cup(2,+\infty)
    $\n<DISAMBIGUATION>Set notation. \infty is infinity, not a variable. \cup is
    union.</DISAMBIGUATION>\n\boxed{$B=(-\infty,0)\cup(2,+\infty)$}
```

# G. More Experimental Results

## G.1. Additional Qualitative Examples

Fig. 10 provides complementary cases illustrating the observed failure modes. We observe that the baseline frequently (i) breaks row/column alignment in complex 2D layouts, leading to structurally invalid renderings; (ii) exhibits long-sequence instability, where attention drift causes missing symbols/substructures in lengthy expressions; (iii) fails at context-aware symbol disambiguation, confusing visually similar tokens (e.g., W vs. $\omega$) when disciplinary context is required; and (iv) is vulnerable to handwriting artefacts, where spurious marks trigger incorrect schema activation (e.g., mis-parsing a single-line expression as multi-line). In contrast, SASR yields more globally consistent layouts and preserves structure under noise/artefacts, resulting in fewer alignment breaks, fewer omissions, and more context-consistent symbol choices.

## G.2. Qualitative Failure Mode Analysis

Figure 11 provides additional qualitative failure cases. These examples are used as qualitative evidence rather than a separately quantified failure-type benchmark.

The cases of unstable global 2D structure and local symbol correctness without global semantic validity are generated by Qwen3-VL-8B-Instruct, which is selected because it is one of the strongest evaluated open-source models and its outputs are representative and interpretable. Similar failure patterns are also observed in other models. In particular, locally plausible

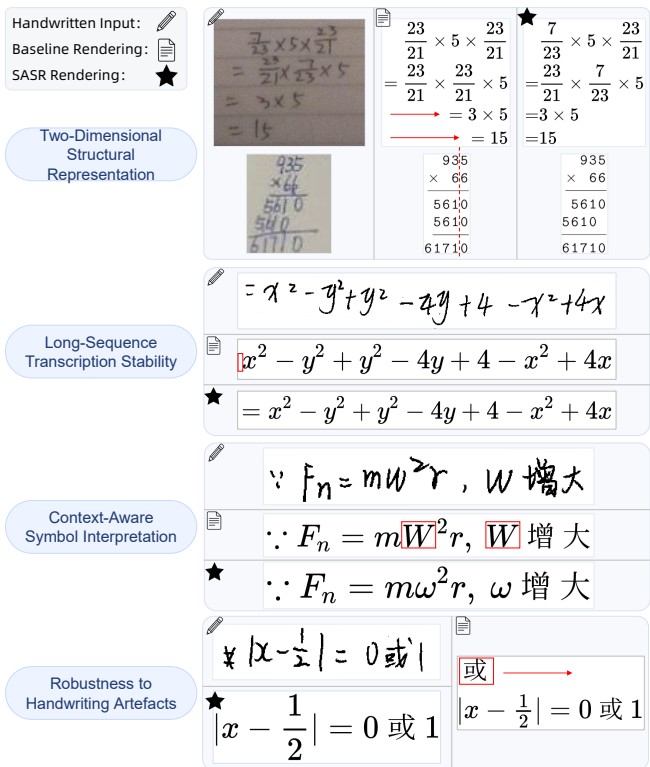

*Figure 10.* **Additional qualitative comparisons on HMER-Bench.** We show handwritten inputs and the corresponding renderings from the baseline and SASR. The examples highlight typical failures of VLM transcription: (i) row/column misalignment in complex 2D layouts, (ii) symbol/substructure omissions under long-sequence decoding, (iii) context-insensitive confusion between visually similar symbols, and (iv) artefact-induced schema flips (e.g., single-line parsed as multi-line) and format violations. Red marks indicate the erroneous positions in baseline outputs.

but globally invalid predictions are common in Nemotron-Nano-12B-v2-VL, while they occur less frequently in stronger models such as Qwen3-VL-8B-Instruct and GLM-4.6v.

As shown in Figure 11, current models often fail to maintain stable global 2D structures in complex expressions. Although local symbols may be correctly recognized, the final output can contain broken alignment, invalid row–column organization, or incorrect structural delimiters. These examples suggest that many HMER errors are not purely caused by local visual recognition, but by unstable global structural parsing during decoding.

### G.3. Failure Analysis on Reduction and Cancellation Expressions

Figure 12 shows representative failure cases in the RCE category. The examples are generated by Qwen3-VL-8B-Instruct and GLM-4.6v, and similar errors are also observed in weaker models.

RCE is challenging because strikethroughs are not meaningless noise, but carry explicit cancellation semantics. Our analysis of RCE predictions from stronger models suggests three main causes: (1) models often treat strikethroughs as meaningless noise and ignore their semantic role in cancellation; (2) even when they recognize their relevance, they often misread the crossed-out symbols because of visual interference; and (3) they may recognize individual symbols but fail to interpret the crossed-out pair as a complete operational meaning, thus missing the reduced result. This suggests that RCE is not merely a robustness issue, but a harder case involving coupled structural semantics and visual interference.

### G.4. Additional Quantitative Results

As shown in the Table 11, Score@CDM, a symbol-level visual-equivalence measure, is consistently much higher than the expression-level ExpRate@CDM across models. For instance, Qwen3-VL-8B achieves an average Score@CDM of 84.17, yet only 25.84 in average ExpRate@CDM. This gap indicates that current VLMs can often recognize local symbols

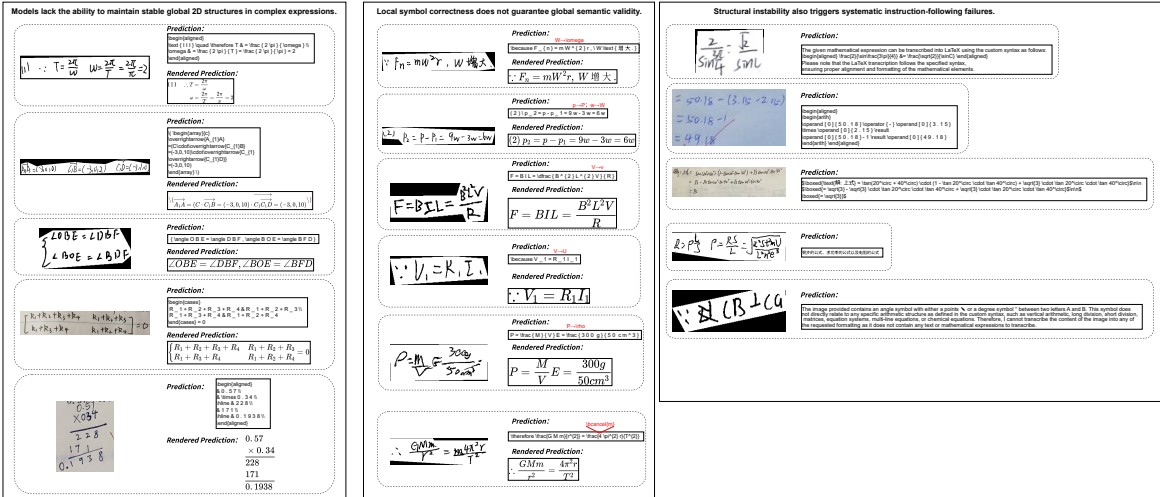

*Figure 11.* Additional qualitative failure cases on HMER-Bench. The examples illustrate unstable global 2D structure, local symbol correctness without global semantic validity, and instruction-following failures caused by structural instability.

reasonably well, but their predictions frequently collapse when global structure, alignment, and semantic consistency must be satisfied simultaneously. The discrepancy is particularly pronounced for structure-intensive categories (e.g., MLD and SOE), where models exhibit a clear "high Score, low ExpRate" pattern. After introducing SASR, ExpRate@CDM improves steadily, with gains concentrated in categories that demand reliable multi-line alignment and structural inference, further confirming that the primary bottleneck in complex handwritten expression recognition lies in structure modeling and global-consistency reasoning, rather than merely symbol recognition.

*Table 11.* **Score@CDM(%) on HMER-Bench for categories where CDM-based visual equivalence evaluation is applicable.** Score@CDM is a character-level visual-equivalence score (CDM-based F1), serving as an approximate proxy for symbol recognition accuracy.

| Model | SSE | LSE | ACE | RCE | SAE | CHEM | SLD | MLD | SOE | MAD | CME | Avg. |
|---|---|---|---|---|---|---|---|---|---|---|---|---|
| SimpleTex | 88.39 | 90.84 | 75.07 | 76.34 | 81.58 | 82.12 | 93.39 | 94.18 | 91.51 | 87.22 | 77.49 | 85.47 |
| Qwen-VL-OCR | 87.45 | 86.90 | 79.04 | 70.62 | 83.16 | 85.89 | 90.32 | 95.05 | 89.52 | 78.86 | 78.29 | 84.59 |
| GOT-OCR2.0 | 78.36 | 65.38 | 58.50 | 62.69 | 64.61 | 56.92 | 70.48 | 67.85 | 53.00 | 57.06 | 49.46 | 61.96 |
| PaddleOCR-VL | 85.97 | 81.51 | 71.85 | 72.44 | 81.40 | 79.98 | 89.79 | 68.79 | 86.62 | 89.76 | 76.43 | 80.79 |
| HunyuanOCR | 90.43 | 92.34 | 78.96 | 77.97 | 85.68 | 89.01 | 95.84 | 91.44 | 93.23 | 83.32 | 82.81 | 87.71 |
| dots.ocr | 67.30 | 69.65 | 51.78 | 47.05 | 56.25 | 0.00 | 59.37 | 26.55 | 0.00 | 29.41 | 0.00 | 36.13 |
| DeepSeek-OCR | 66.32 | 67.20 | 53.56 | 55.02 | 64.64 | 70.03 | 76.10 | 69.07 | 62.58 | 62.01 | 55.22 | 63.99 |
| MonkeyOCR | 89.98 | 91.62 | 79.35 | 56.62 | 84.23 | 0.16 | 94.49 | 93.83 | 0.00 | 80.76 | 0.19 | 60.02 |
| Chandra | 56.13 | 72.29 | 51.84 | 53.44 | 64.75 | 67.09 | 76.98 | 73.51 | 75.27 | 83.96 | 63.81 | 67.60 |
| Doubao-Seed-1.6 | 86.42 | 90.75 | 76.47 | 70.37 | 87.47 | 83.26 | 92.04 | 90.93 | 87.70 | 89.31 | 81.35 | 85.67 |
| Phi-4-multimodal-instruct | 53.33 | 49.02 | 38.62 | 45.79 | 55.10 | 50.47 | 53.57 | 50.41 | 50.08 | 36.40 | 37.52 | 47.21 |
| Ovis2.5-9B | 83.44 | 83.75 | 66.76 | 62.23 | 79.72 | 81.16 | 83.27 | 61.27 | 87.98 | 82.89 | 71.28 | 77.38 |
| Nemotron-Nano-12B-v2-VL | 81.66 | 84.25 | 67.95 | 72.30 | 77.66 | 78.47 | 88.93 | 52.79 | 73.41 | 70.83 | 67.59 | 74.31 |
| Llama 4 Maverick | 70.15 | 73.61 | 57.08 | 53.70 | 75.22 | 78.71 | 80.37 | 51.68 | 83.71 | 79.11 | 69.21 | 71.11 |
| Qwen3-VL-235B-Instruct | 92.16 | 93.94 | 81.35 | 79.42 | 90.65 | 89.89 | 95.36 | 95.56 | 92.96 | 91.15 | 85.69 | 90.19 |
| Qwen3-VL-8B-Instruct | 87.81 | 87.72 | 74.43 | 66.95 | 87.61 | 86.14 | 90.49 | 78.00 | 90.45 | 88.21 | 79.74 | 84.17 |
| Qwen3-VL-8B-Instruct(ours) | 90.80 | 91.69 | 79.21 | 76.95 | 87.31 | 89.56 | 93.47 | 95.37 | 91.92 | 87.35 | 82.77 | 88.21 |

**Detailed robustness results.** Table 12 reports ExpRate for each noise standard deviation $\sigma$. SASR preserves strong performance as noise increases (23.53 at $\sigma=0$ to 21.60 at $\sigma=30$), while the baseline remains near a low-accuracy regime (10.62 to 10.04) and shows non-monotonic fluctuations. Across all $\sigma$, SASR maintains a consistent absolute margin over the baseline (min gap: $21.60 - 9.50 = 12.10$ points), and achieves a smaller relative drop (9.36% vs. 15.78%; computed as

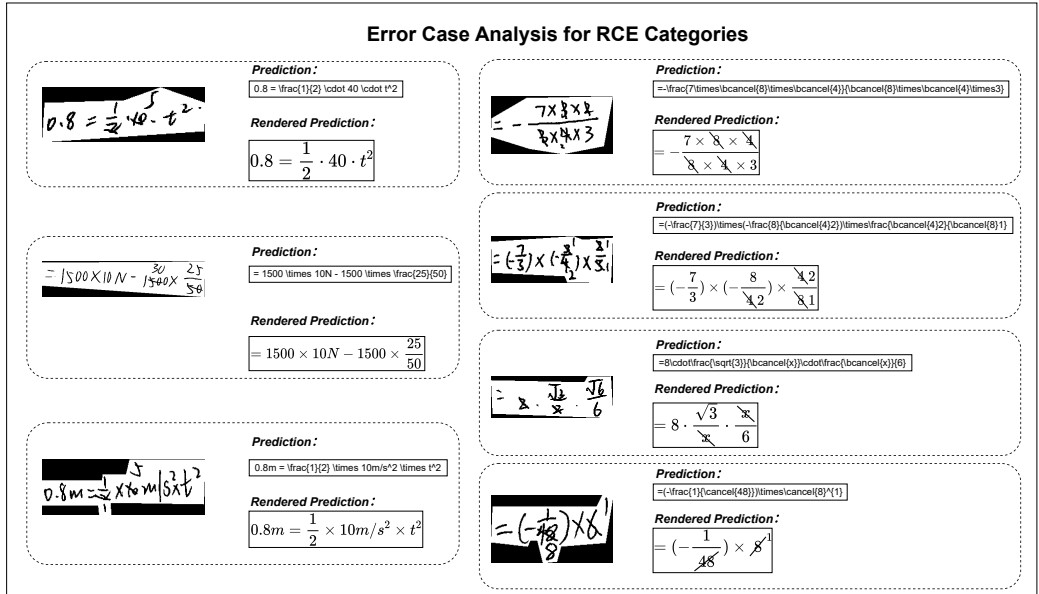

*Figure 12.* Representative failure cases in the RCE category. Models may ignore strikethroughs, misread crossed-out symbols, or fail to interpret cancellation as an operation.

$(\max - \min)/\max$ over all $\sigma$). These results support that SASR improves performance retention under visual perturbations beyond a baseline floor effect.

| $\sigma$ | 0 | 3 | 5 | 8 | 10 | 15 | 20 | 25 | 30 |
|---|---|---|---|---|---|---|---|---|---|
| Baseline | 10.62 | 11.28 | 10.69 | 10.76 | 10.17 | 10.76 | 9.50 | 9.58 | 10.04 |
| SASR (Ours) | **23.53** | **23.46** | **23.79** | **23.39** | **23.83** | **23.31** | **22.49** | **22.12** | **21.60** |

*Table 12.* Per-level robustness of Qwen3-VL-8B-Instruct to zero-mean Gaussian noise on the 8-bit intensity scale (pixel values clipped to [0,255], ExpRate,%).

**Detailed Generalization Results.** Table 13 reports per-category ExpRate(%) on the 13 HMER-Bench categories after transferring the same SASR inference pipeline to Ovis2.5-9B and Nemotron-Nano-12B-v2-VL. SASR yields consistent gains on both models: the average ExpRate of Ovis improves from 7.72 to 12.55 (+4.83, +62.6%), and Nemotron improves from 6.01 to 8.91 (+2.90, +48.3%). The improvements are most pronounced on categories that are more sensitive to 2D structure and cross-line alignment (D2). On Ovis, SDE (short division) shows the largest gain (13.11→36.07, +22.96), with clear improvements also on MLD (0.00→9.80, +9.80), SOE (18.58→30.09, +11.51), VAE (15.69→21.57, +5.88), and MAD (19.81→26.42, +6.61). On Nemotron, structural categories likewise account for most of the gain, e.g., SOE (0.88→9.73, +8.85) and MAD (0.94→10.38, +9.44), alongside steady improvements on SDE (0.00→4.91) and VAE (0.00→3.92). In contrast, some categories remain challenging (e.g., RCE stays at 0.00 on both models), suggesting that the dominant bottleneck lies in complex-structure modeling and compliance with normalized output conventions rather than purely character-level recognition.

*Table 13.* Per-category ExpRate(%) of SASR generalization on HMER-Bench when transferring the same inference pipeline to Ovis2.5-9B and Nemotron-Nano-12B-v2-VL.

| Model | SSE | LSE | ACE | RCE | SAE | CHEM | SLD | MLD | SDE | VAE | SOE | MAD | CME | Avg. |
|---|---|---|---|---|---|---|---|---|---|---|---|---|---|---|
| Ovis2.5-9B(baseline) | 15.79 | 1.89 | 1.71 | 0.00 | 7.83 | 3.74 | 0.86 | 0.00 | 13.11 | 15.69 | 18.58 | 19.81 | 1.47 | 7.72 |
| Ovis2.5-9B(SASR) | 16.67 | 2.83 | 5.13 | 0.00 | 8.70 | 6.54 | 2.59 | 9.80 | 36.07 | 21.57 | 30.09 | 26.42 | 3.68 | 12.55 |
| Nemotron-Nano-12B-v2-VL(baseline) | 27.19 | 12.26 | 0.85 | 0.00 | 10.43 | 0.00 | 16.38 | 0.00 | 0.00 | 0.00 | 0.88 | 0.94 | 2.21 | 6.01 |
| Nemotron-Nano-12B-v2-VL(SASR) | 30.70 | 12.26 | 1.71 | 0.00 | 11.30 | 2.80 | 17.24 | 1.96 | 4.91 | 3.92 | 9.73 | 10.38 | 2.21 | 8.91 |

