# OpenReview forum: "Seeing Symbols, Missing Structure: A Real-World Handwritten Mathematical Expression Recognition Benchmark for Large Models"
_ICML.cc/2026/Conference — ICML 2026 regular_

### Official Review · Reviewer_HT1G · 2026-03-10

**Soundness:** 3
**Presentation:** 3
**Significance:** 2
**Originality:** 3
**Overall Recommendation:** 4
**Confidence:** 4

**Summary:**

This paper focuses on handwritten mathematical expression recognition in real educational scenarios. The authors introduce HMER-Bench, a benchmark with realistic handwriting artifacts and structurally complex expressions. The authors further propose a training-free structure-aware inference framework that improves ExpRate on several models. Experiments demonstrate the effectiveness of the proposed method.

**Compliance With Llm Reviewing Policy:**

Affirmed.

**Final Justification:**

The rebuttal addressed my concerns, so I maintain my score unchanged.

**Key Questions For Authors:**

1. Could the authors provide further discussion and comparison with M2E, another benchmark for two-dimensional mathematical problems, including some visual comparisons?

2. Can the authors show that the proposed method improves on other public HMER datasets that were not used in constructing this benchmark?

3. The benchmarks proposed in the paper are all single formulas, while the methods compared in the experimental tables are all for document-level scenarios. Is there a potential gap in such a direct comparison?

**Limitations:**

yes

**Strengths And Weaknesses:**

Strengths:
1. This paper focuses on solving a problem of great practical value. The authors propose a training-free, schema-anchored structure-aware inference framework, which achieves promising performance.

2. This paper is well-written and easy to understand.


Weaknesses:
1. M2E is also a benchmark related to two-dimensional mathematical problems. This paper lacks further discussion and comparison with it.  It would be better if more comparisons, including some visual ones with M2E, could be included.

2. The authors state: "Our results show that structural inference, rather than symbol recognition, is the primary bottleneck of models." However, the experiment is insufficiently conducted. They only tested the proposed method on the proposed benchmark. The popular benchmarks such as CROHME and M2E are lacking.

3. The authors only verified the effectiveness of the proposed method on Qwen3-VL-8B. However, they claim to have proposed an inference framework. Therefore, it would be better to verify the effectiveness of more methods on this framework.

---

> ### Author Rebuttal · Authors · 2026-03-31
>
> Thank you for the valuable comments. We respond to them below.
>
> **On comparison with M2E.**
> We will add a clearer comparison with M2E in the revision, together with a visual example (**Image[https://bashify.io/i/SYixuo ]**). The key difference is not simply whether multi-line expressions are included, but what is annotated and therefore what can be rigorously evaluated. M2E supports two-dimensional handwritten math, but its annotation target is closer to line-level or segmented recognition, which is useful for multi-line content yet does not explicitly preserve cross-line correspondence, row-column alignment, or whole-structure semantics. In contrast, HMER-Bench is designed for overall 2D structure recovery: it uses unified, renderable LaTeX annotations that preserve the full structure and alignment semantics of categories such as multi-line equations, vertical arithmetic, short division, matrices/determinants, and systems of equations. We will add both a textual comparison and a more informative visual example to make this distinction explicit.
>
> **On the claim about structural bottlenecks.**
> Our claim is not that structural inference is the primary bottleneck on all HMER datasets, but specifically on the real educational distribution covered by HMER-Bench. We will revise the wording to make this scope explicit. Existing benchmarks such as CROHME and M2E emphasize different aspects of HMER. CROHME is dominated by single-line expressions, where the structural search space is much smaller. M2E includes multi-line expressions, but its annotations are closer to line-wise decomposition and do not explicitly preserve the global structure and alignment semantics central to our setting. By contrast, HMER-Bench is designed to complement these benchmarks by explicitly covering structure-intensive educational cases, including complex 2D layouts, alignment, vertical arithmetic, short division, matrices/determinants, and systems of equations (**Image[https://bashify.io/i/6jrTMG ]**). Since SASR first infers a structural schema and then constrains recognition accordingly, it mainly benefits settings where global 2D structure is present and evaluation-relevant. We will narrow the claim accordingly.
>
> **On validation across more models.**
> Section 5.5 already applies SASR to Ovis2.5-8B and Nemotron-Nano-12B-v2-VL, with consistent improvements. We have further added experiments on GLM-4.6v and KIMI-K2.5; full results are available at **Image[https://bashify.io/i/Rge8hW ]**. These results support that SASR is transferable across model families, although the gain magnitude varies with the base model’s visual, structural, and instruction-following capabilities. We will include this point more explicitly in the revision.
> | Method              | Avg.ExpRate   |
> | ------------------- | ------------- |
> | KIMI-K2.5(baseline) | 28.06         |
> | KIMI-K2.5(SASR)     | 38.90(+10.84) |
> | GLM-4.6v(baseline)  | 13.88         |
> | GLM-4.6v(SASR)      | 25.24(+11.36) |
>
> **On possible mismatch between benchmark and compared methods.**
> The purpose of these comparisons is not to argue that single-formula recognition is inherently best handled by document-level models, but to assess how currently available large models and OCR/MER systems perform on reconstructing real handwritten formulas in practice. To reduce bias, for models that support custom prompts, we provide unified annotation guidelines and output constraints; for OCR-oriented VLMs and MER APIs that do not support prompt control, we do not impose extra inference constraints. In addition, models that cannot support custom prompts are scored as 0 in categories such as VAE/SDE that rely on custom annotations, and are excluded from the average. Therefore, these results should be interpreted as an assessment of practical real-world capability rather than a comparison under fully idealized settings. We will clarify this more explicitly.
>
> If there are still any concerns, we would be happy to further clarify them.

---

> > ### Author Rebuttal · Reviewer_HT1G · 2026-04-03
> >
> > The rebuttal addressed my concerns, so I maintain my score unchanged.

---

### Official Review · Reviewer_grLC · 2026-03-10

**Soundness:** 3
**Presentation:** 3
**Significance:** 3
**Originality:** 2
**Overall Recommendation:** 4
**Confidence:** 3

**Summary:**

This paper proposes a Handwritten Mathematical Expression Recognition (HMER) benchmark and a training-free framework to bridge the structural reasoning gap in real-world educational scenarios. The benchmark comprises 13,513 real-world educational samples across 13 categories under four diagnostic dimensions, designed to identify recurring structural failure patterns in existing models. A key empirical observation motivates both contributions: despite achieving symbol-level accuracy of ~ 84% (Score@CDM), existing models reach only ~ 26% expression-level accuracy (ExpRate@CDM), indicating that structural reasoning rather than symbol recognition is the primary bottleneck. To address this, the paper proposes a training-free, multi-stage structure-aware inference framework leveraging an existing VLM, which improves ExpRate from 11.63% to 24.52% on Qwen3-VL-8B without parameter updates

**Compliance With Llm Reviewing Policy:**

Affirmed.

**Ethical Review Flag:**

Flag this paper for an ethics review.

**Final Justification:**

The rebuttal improves clarity and helps clarify the contributions. However, some limitations remain regarding quantitative evidence and broader comparisons, so I keep my original evaluation.

**Key Questions For Authors:**

1. The paper evaluates SASR primarily on HMER-Bench. Could the authors demonstrate generalization by applying SASR to existing benchmarks (e.g., HME100K, CROHME) or additional model architectures? A positive result would substantially strengthen the methodological contribution.
2. Could the authors clarify the rationale for selecting Qwen3-VL-8B, including which pre-trained weights were used and whether the observed gains are specific to this model's pretraining? This would help assess whether SASR is broadly applicable or model-dependent.
3. The six evaluation axes in Table 1 appear to be author-defined. Could the authors clarify whether these criteria are adopted from prior work or newly proposed? If newly proposed, what justifies treating them as independent axes, particularly given the potential overlap between 2D and Align?
4. The four diagnostic dimensions also appear to be author-defined. Is there a theoretical or empirical basis for why these four dimensions are necessary and sufficient to characterize structural reasoning failures in HMER? Additionally, are D1 (linear coherence) and D2 (2D structure) sufficiently distinct to warrant separate dimensions?

**Limitations:**

Yes.

**Strengths And Weaknesses:**

Strengths
1. Well-supported motivation. The gap between Score@CDM (84%) and ExpRate@CDM (26%) clearly supports the central claim that structural reasoning, rather than symbol recognition, is the main bottleneck in modern HMER systems.
2. Comprehensive benchmark contribution. HMER-Bench combines real educational data, a structured diagnostic taxonomy, and arithstudio.sty for arithmetic layouts that previously lacked a unified format. Both the dataset and the representation standard offer independent value to the community.
3. Practical and effective framework. SASR requires no additional training data or parameter updates, achieves +12.89 ExpRate improvement on Qwen3-VL-8B, and shows consistent gains across two additional models.

Weaknesses
1. Limited methodological contribution. While the theoretical framework in Appendix A is mathematically sound, the implementation relies on structured prompting and in-context learning — both well-established techniques. It remains unclear whether the observed gains stem from structural schema conditioning or general chain-of-thought effects, as these are not empirically isolated.
2. Cross-model performance differences are not discussed. The paper focuses primarily on Qwen3-VL-8B without justifying this choice. The notable drop in gains across other models (Qwen: +12.89, Ovis: +4.83, Nemotron: +2.90) is not analyzed, leaving the generalizability of SASR an open question.
3. The benchmark scale may affect evaluation reliability. With approximately 1,000 samples per category (13,513 total), the benchmark is relatively modest compared to existing datasets such as HME100K (~100K). While the focus on structural diversity and real handwriting is a clear strength, the limited scale warrants consideration regarding statistical robustness.
4 (minor). Terminology in Table 1 needs clarification. The distinction between 2D structural coverage and Align is not clearly explained within the paper, making the dataset comparison somewhat difficult to interpret.

---

> ### Author Rebuttal · Authors · 2026-03-31
>
> Thank you for the valuable comments. We respond to them below.
>
> **On the methodological contribution of SASR.**
> The main contribution of this work is not only SASR, but also HMER-Bench itself. Specifically, the paper contributes: (1) HMER-Bench, a diagnostic benchmark that exposes the structural weaknesses of current models on real educational handwritten mathematical expressions, and (2) SASR, a training-free, interpretable, and transferable test-time framework designed to address the failures revealed by this benchmark. SASR should therefore be understood together with the benchmark: the benchmark identifies the problem setting, and SASR provides a lightweight solution. The observed gains are also not merely a generic CoT effect. As shown in Sec. 5.4, removing both MSR and SIL gives 11.63 Avg. ExpRate; adding only MSR raises this to 19.52 (+7.89); adding only SIL gives 14.02 (+2.39); and combining both reaches 24.52 (+12.89). This indicates that the improvement comes from two complementary mechanisms: MSR reduces structural uncertainty through schema-consistent constrained generation, while SIL stabilizes schema inference through anchor examples.
>
> **On cross-model differences.**
> We chose Qwen3-VL-8B because it is strong among open-source models of similar scale and fully reproducible. As stated in Sec. 5.1, SASR is applied directly to its public pretrained weights at test time, with no retraining or parameter updates. We agree that cross-model analysis should be clearer. Sec. 5.5 already shows transfer to Ovis2.5-8B and Nemotron-Nano-12B-v2-VL, improving Avg. ExpRate from 7.72 to 12.55 and 6.01 to 8.91. We further added GLM-4.6v and KIMI-K2.5, which also show clear gains: **Image [https://bashify.io/i/Rge8hW ]**
>
> | Method              | Avg.ExpRate   |
> | ------------------- | ------------- |
> | KIMI-K2.5(baseline) | 28.06         |
> | KIMI-K2.5(SASR)     | 38.90(+10.84) |
> | GLM-4.6v(baseline)  | 13.88         |
> | GLM-4.6v(SASR)      | 25.24(+11.36) |
>
> Overall, SASR is transferable, while the absolute gain depends on the base model’s visual resolution, instruction-following, and structural representation ability.
>
> **On benchmark scale, reliability, and Table 1 terminology.**
> HMER-Bench is not intended to be a large general-purpose training corpus like HME100K or MathWriting. Instead, it is a diagnostic benchmark for structural failures in real educational handwritten expressions. Its 13,513 samples were deliberately selected to stress complex 2D structure, alignment, arithmetic layout, handwriting artifacts, and real educational distribution shift. For this purpose, a targeted benchmark is more suitable than a much larger corpus not designed for such diagnosis. We will also clarify the terminology in Table 1. In our usage, 2D refers to non-linear layouts beyond single-line expressions, such as multi-line derivations, columnar arithmetic, short division, matrices, and systems of equations. Align is stricter: it refers to whether row-column correspondence and alignment are explicitly preserved for evaluation.
>
> **On broader generalization experiments.**
> HME100K and CROHME are not the main validation targets because they mainly focus on single-line expressions and provide limited coverage of alignment, arithmetic layouts, and handwriting artefacts. Thus, positive results there would not directly validate the failure modes targeted by SASR. Still, SASR consistently improves performance on Qwen3-VL-8B, Ovis2.5-8B, Nemotron-Nano-12B-v2-VL, GLM-4.6v, and KIMI-K2.5, supporting its transferability across model families.
>
> **On the six evaluation axes.**
> The six axes in Table 1 are descriptive criteria proposed in this paper rather than a fixed taxonomy adopted from prior work. We introduced them because existing HMER datasets lack a concise framework for describing structural properties in real educational scenarios. They summarize educational authenticity, 2D structure, alignment, elementary arithmetic expressions, real-world noise, and fine-grained coverage. We do not claim that they are fully independent; for example, 2D and Align are related. They are intended as a practical comparison framework rather than a complete taxonomy.
>
> **On the four diagnostic dimensions.**
> Similarly, D1–D4 are practical diagnostic dimensions derived from error patterns observed on HMER-Bench; we do not claim that they form a necessary-and-sufficient theoretical decomposition of HMER reasoning failures. D1 focuses on linear structural coherence, such as omissions, repetitions, and ordering errors in longer single-line expressions, while D2 focuses on non-linear 2D structure reconstruction, including hierarchical layout, cross-row relations, row-column alignment, and implicit grids. We will clarify this distinction and avoid wording that may imply a complete theoretical taxonomy.
>
> If there are still any concerns, we would be happy to further clarify them.

---

> > ### Author Rebuttal · Reviewer_grLC · 2026-04-02
> >
> > If I understand correctly, the main contribution lies in the benchmark. Although the method itself has limited novelty and builds on existing prompting-based techniques, the overall contribution remains acceptable.

---

> > > ### Author Response · Authors · 2026-04-08
> > >
> > > Thank you for the positive assessment and for noting that the rebuttal improved the paper’s clarity. We appreciate your overall supportive evaluation. We would just like to add a brief clarification on the point about quantitative evidence and broader comparisons, so that this part of the paper is reflected as clearly as possible in the final discussion.
> > >
> > > Specifically, both aspects were already included in the main paper and were further supplemented in the rebuttal.
> > >
> > > In the main paper, we already provide:
> > > (1) a benchmark-scale evaluation on HMER-Bench (13,513 samples, 13 categories, and 4 diagnostic dimensions);
> > > (2) clear quantitative gains of SASR on Qwen3-VL-8B (Avg. ExpRate 11.63% → 24.52%);
> > > (3) an ablation showing that the improvement is not simply due to generic CoT prompting, since MSR only achieves 19.52%, SIL only 14.02%, and MSR+SIL 24.52%;
> > > (4) cross-model transfer results on Ovis2.5-8B (7.72% → 12.55%) and Nemotron-Nano-12B-v2-VL (6.01% → 8.91%); and
> > > (5) robustness results under Gaussian noise, where SASR exhibits a smaller relative drop than the baseline.
> > >
> > > In the rebuttal, we further supplemented the comparison scope by adding a specialized HMER baseline, Uni-MuMER, under the same transfer-without-retraining setting. We also added results on two additional model families, GLM-4.6v (13.88% → 25.24%) and KIMI-K2.5 (28.06% → 38.90%), to further illustrate the transferability of SASR.
> > >
> > > We hope this brief clarification is helpful, and thank you again for your careful reading and constructive feedback.

---

### Official Review · Reviewer_JnhP · 2026-03-12

**Soundness:** 3
**Presentation:** 3
**Significance:** 2
**Originality:** 2
**Overall Recommendation:** 4
**Confidence:** 5

**Summary:**

This paper introduces HMER-Bench, a real-world benchmark comprising 13,513 handwritten mathematical expression samples organized into 13 fine-grained categories spanning four diagnostic dimensions. The data is collected from authentic Chinese educational examination and homework sources, covering expressions with diverse complexity levels including matrices, systems of equations, vertical arithmetic, short division, and mixed expressions. The paper also proposes a training-free Schema-Anchored Structure-Aware Reasoning (SASR) method, which decomposes HMER into a multi-stage process of structural schema inference (identifying the expression's global layout type), schema-constrained transcription, context-driven symbol disambiguation, and final structured output.

**Compliance With Llm Reviewing Policy:**

Affirmed.

**Final Justification:**

I thank the authors for their rebuttal and the additional experiments. Although the rebuttal has not fully resolved all of my original concerns, after reading the other reviewers' comments and taking a broader view of this work's reference value to the HMER community, I have decided to raise my recommendation to weak accept.

**Key Questions For Authors:**

1. How many unique annotators participated in labeling? What was the inter-annotator agreement (e.g., on LaTeX correctness vs. structural category assignments)?

2. What is the underlying reason for RCE categories registering exactly 0.00% across all models?

3. Writing Issue: The acronym "HEMR" appears on page 3 of the Related Work section likely a typo for "HMER" and should be corrected.

**Limitations:**

- The benchmark is primarily sourced from Chinese educational materials, limiting cross-cultural generalizability.

-- The training-free SASR framework depends entirely on prompt design and model instruction-following capability, making it brittle for models that do not support custom system prompts.

**Strengths And Weaknesses:**

Strengths:
--  MER-Bench is a new useful benchmark for MHE.  Most existing benchmarks (CROHME, HME100K) focus on simple single-line expressions. The four-dimensional diagnostic taxonomy systematically links observable transcription failures to structural root causes, enabling principled capability attribution that is absent in prior work.

 -- The paper evaluates 15+ models spanning general-purpose VLMs, OCR-tuned VLMs, and specialized MER APIs, providing a broad view of the current state of the field.

 -- SASR introduces a principled probabilistic decomposition (Eq. 1–5) that separates structural schema inference from symbolic transcription.

-- The authors develop a custom LaTeX annotation style (`arithstudio.sty`) to handle primary-school-specific layouts (vertical arithmetic, short division) for which no prior unified annotation standard existed.


Weaknesses:
1. The core operation of SASR decomposes a complex task into ordered Chain-of-Thought (CoT) steps and stabilizes classifier outputs via in-context exemplars, this is the standard paradigm of Prompt Engineering. While the theoretical justification provided in Appendix offers certain formal value, it cannot obscure the fact that SASR is essentially a carefully designed prompt chain. Compared to explicit structural encoding approaches such as Tree-CoT (Uni-MuMER) or PosFormer, this method exhibits a clear gap in technical depth. Given that ICML prioritizes methodological innovation as its core criterion, this constitutes the most fundamental issue facing by this paper.

2. The paper evaluates VLMs and OCR APIs, but provides limited evaluation of traditional encoder-decoder HMER systems (e.g., CoMER, SAN, BPD, ...) that were specifically designed for structural reasoning. Including these would provide context for how much progress VLMs have made relative to the prior dedicated HMER literature, and whether VLMs actually represent an improvement despite their structural weaknesses.

3. Appendix A presents four propositions/theorems, including the chain decomposition of the total probability formula (Prop. 1–2), the non-negativity of mutual information (Thm. 1), and an upper bound on MAP approximation error (Thm. 2). These are all foundational results in probability theory, with proofs derived entirely from textbook-level definitions. The paper packages these known results as "theoretical justification," offering no substantive insight into SASR's working mechanism; rather, it may create an illusion of theoretical depth for readers.

4.  ASR achieves an 70.20% ExpRate on SDE using Qwen3-VL-8B, compared to only 17.43% for the baseline, a remarkable improvement primarily driven by the inclusion of complete usage instructions for arithstudio.sty in the system prompt, essentially telling the model how to format the correct output. This suggests that the performance gains on SDE stem largely from annotation format guidance rather than genuine improvements in structural reasoning capabilities. These two aspects should have been discussed separately from other categories of improvement. However, the paper presents them collectively as "SASR's structural reasoning gains," creating ambiguity in the argumentation.

5.

---

> ### Author Rebuttal · Authors · 2026-03-31
>
> Thank you for the valuable comments. We respond to them below.
>
> **On the methodological nature of SASR.**
> SASR is not an architecture-heavy method like PosFormer or Uni-MuMER, and we will clarify this more explicitly. Our contribution is not a new foundation model, but a benchmark-driven test-time framework tailored to the structural failures revealed by HMER-Bench. Specifically, the paper contributes: (1) HMER-Bench, which exposes the structural weaknesses of current models on complex handwritten mathematical expressions, and (2) SASR, a lightweight, interpretable, training-free framework that turns these findings into a practical improvement strategy through explicit structure-aware decomposition. While its high-level form is related to prompt engineering, SASR is not a generic prompt chain: its decomposition, stage ordering, and constraints are specifically designed for mathematical expression recognition and target ambiguities such as 2D layout, alignment, arithmetic formats, and handwriting artifacts. We will position SASR more precisely as a low-cost, domain-specialized framework.
>
> **On comparison with dedicated HMER systems.**
> Traditional HMER systems do provide useful context. However, HMER-Bench is not intended to measure how well supervised HMER systems fit a benchmark-specific training distribution, but whether current large models can recognize real educational expressions without task-specific retraining. Including encoder-decoder HMER systems such as CoMER, SAN, or BPD in a fully comparable way would require benchmark-specific splits and retraining, which would shift the setting from out-of-domain structural generalization to supervised fitting. To provide partial context, we added Uni-MuMER because reproducible public weights are available. Its average ExpRate on HMER-Bench is only 11.48% (D1: 33.11%, D2: 6.79%, D3: 8.56%, D4: 8.83%), indicating that strong performance on standard HMER benchmarks does not transfer well to the more structurally complex setting studied here.
>
> **On the theoretical justification.**
> Appendix A is not intended to present new theoretical results. Its purpose is only to formalize the intuition behind SASR: HMER can be viewed as schema inference followed by schema-conditioned transcription, and test-time structural constraints can reduce ambiguity in this decomposition. The appendix is included only to make the design rationale more explicit and self-contained, not to claim theoretical novelty.
>
> **On possible confounding from format instruction on SDE.**
> This factor should indeed be discussed separately. Importantly, the baseline is also provided with the full usage instruction for `arithstudio.sty`; thus, the comparison is not between a model without format guidance and one with it. What SASR additionally introduces is explicit schema identification, ordered reasoning, and constraint-based transcription. In other words, the style instruction addresses how to format the output, while SASR addresses what structure is present and how to transcribe it more reliably. We will separate format-guidance effects from broader structure-aware improvements.
>
> **On annotation transparency.**
> HMER-Bench was annotated with the support of a professional annotation company. The full process covered about 140,000 samples, from which we selected 13,513 representative images for the final benchmark. Reliability was ensured through standardized guidelines, company-side quality control, and two rounds of full manual verification by our team, with mandatory re-annotation for non-compliant samples. Under a strict criterion where even a single-character error counts as incorrect, the pass rate reached 99% after the second round. We also normalized labels (e.g., `\ne/\neq`, exponent notation) to reduce non-essential inconsistencies.
>
> **On the zero performance on RCE.**
> As shown in **Figure [https://bashify.io/i/7cnPWi ]**, our analysis of RCE predictions from stronger models suggests three main causes: (1) models often treat strikethroughs as meaningless noise and ignore their semantic role in cancellation; (2) even when they recognize their relevance, they often misread the crossed-out symbols because of visual interference; and (3) they may recognize individual symbols but fail to interpret the crossed-out pair as a complete operational meaning, thus missing the reduced result. This suggests that RCE is not merely a robustness issue, but a harder case involving coupled structural semantics and visual interference.
>
> **On limitations.**
> We acknowledge both limitations. First, the benchmark is mainly sourced from Chinese educational materials, which limits cross-cultural generalizability. Second, as a training-free method, SASR depends on prompt design and the instruction-following ability of the underlying model, making it less suitable for models that do not support custom system prompts well.
>
> If there are still any concerns, we would be happy to further clarify them.

---

### Official Review · Reviewer_PJ4H · 2026-03-13

**Soundness:** 2
**Presentation:** 2
**Significance:** 3
**Originality:** 3
**Overall Recommendation:** 4
**Confidence:** 4

**Summary:**

The paper proposes HMER-Bench, a benchmark for handwritten mathematical expression recognition (HMER), focusing on multi-line expression coming from real-world educational sources, notably including short division and other rigidly structured samples. It additionally proposes a method called Schema-Anchored Structure-aware Reasoning (SASR) to improve the ability of Qwen3-8B to perform HMER through the use of in-context-learning and structure-aware reasoning. On HMER-Bench, the proposed method performs better than all tested VLM baselines. The benchmark consists of 13 categories, and evaluation is reported for each category, and as an average.

**Compliance With Llm Reviewing Policy:**

Affirmed.

**Final Justification:**

My concerns have been addressed; therefore, I raise my score. I thank the authors for the clarifications and look forward to seeing the revised manuscript with the requested additions.

**Key Questions For Authors:**

1. How do existing HMER models perform on your benchmark?
2. Can you provide experimental results backup for the failure mode analysis in Section 5.3?
3. What is the equivalence threshold for ExpRate@CDM used in this paper and mentioned in Section 5.1?

**Limitations:**

yes (unless one considers the weaknesses pointed out)

**Strengths And Weaknesses:**

Strengths:

- The paper contains an ample selection of OCR-oriented and general-purpose VLMs that have been tested on the proposed benchmark, which is very challenging for recent VLMs, and the split evaluation allows for granular evaluation of the capability of models to recognize different kinds of expressions
- The proposed method improves with respect to its baseline when applied to Qwen, Ovis, and Nemotron

Weaknesses:

- None of the tested approaches other than SASR are specialized for HMER, except for SimpleTex.
- No justification for the exclusion of all of the HMER methods mentioned in Section 2 from the experimental evaluation on HMER-Bench is given, in particular, both recent VLM-based methods like Uni-MuMER, HiE-VL or VLPG, and older baselines like CoMER or TAMER. These specialized models are shown in the literature (for example, the cited Wang et al. 2025 and Li et al., 2025b) to outperform even closed-source general-purpose VLMs in this task on other benchmarks.
- The authors heavily rely on a custom LaTeX environment to represent some elements of the benchmark, possibly introducing bias
- Remarks in Section 5.3 and Appendix F.2 that the difference in ExpRate@CDM and Score@CDM is a clear indicator of a structural understanding issue with current models are a bit of overstatements: ExpRate@CDM, especially with a threshold close to 1 in long expressions, is a very punishing score when compared with Score@CDM
- The failure mode analysis in Section 5.3 is not backed up experimentally
- The equivalence threshold for ExpRate@CDM mentioned in Section 5.1 is not specified. The cited Wang et al., 2025 paper uses 1 as an equivalence threshold; on the other hand, the definition given in this paper is vague.
- The inclusion of Score@CDM for SASR only in the appendix is confusing, especially because it shows that SASR also has a wide gap between Score@CDM and ExpRate@CDM, undermining the combination of claims made in the paper that this gap is caused by the inability of models’ structure parsing, and that SASR significantly improves on this.

---

> ### Author Rebuttal · Authors · 2026-03-31
>
> Thank you for the thoughtful comments. We have clarified several previously under-explained points.
>
> **On HMER-specific baselines and omitted methods.**
> We agree that specialized HMER models are important references. However, the goal of HMER-Bench is not to benchmark fully supervised in-domain HMER performance, but to diagnose the limitations of current models in realistic educational scenarios involving complex 2D layouts, alignment-sensitive structures, arithmetic formats, and handwriting artifacts. For this reason, our main comparison emphasizes general-purpose VLMs, OCR-oriented VLMs, and MER/OCR APIs, which are more suitable for evaluating out-of-domain structural generalization without benchmark-specific training.
>
> Following the reviewer’s suggestion, we added Uni-MuMER as a supplementary HMER-specific baseline and will clarify this evaluation scope in the revision. We also agree that HiE-VL, VLPG, CoMER, and TAMER are relevant. Their exclusion was due to evaluation comparability rather than oversight. Uni-MuMER was included because reproducible public weights are available. HiE-VL and VLPG were not included because fair and verifiable reproduction was not feasible due to missing public code, weights, or key implementation details. Earlier supervised HMER methods such as CoMER and TAMER would require benchmark-specific train/validation/test splits and retraining on HMER-Bench, which would change the setting from transfer evaluation without extra training into benchmark-specific supervised fitting. This is inconsistent with the intended role of HMER-Bench as a diagnostic benchmark rather than a new supervised training benchmark.
>
> Importantly, the supplementary Uni-MuMER result supports this distinction: although it is a strong specialized model, it transfers poorly to HMER-Bench, achieving only 11.48% average ExpRate, with correct cases concentrated on relatively simple expressions (D1: 33.11%, D2: 6.79%, D3: 8.56%, D4: 8.83%). We will make this rationale explicit.
>
> **On the custom LaTeX environment.**
> We would like to clarify that the benchmark does not heavily rely on custom LaTeX syntax. The custom environment is used only for a small subset of samples in VAE and SDE, where standard LaTeX cannot consistently represent the target structures; all other categories use standard LaTeX. For models that allow prompt customization, both the baseline and SASR receive exactly the same formatting instruction. Thus, SASR’s gains do not come from privileged access to custom syntax, but from better identifying structural type and transcribing it accordingly. For models without prompt customization, we exclude VAE and SDE from averaged results to avoid unfair comparison.
>
> **On the Score@CDM–ExpRate@CDM gap.**
> We agree that our original wording was too strong. Our intention was not to claim that the Score@CDM–ExpRate@CDM gap alone proves deficient structural understanding. Rather, we view it as a useful diagnostic signal when interpreted together with other evidence. Since CDM already reduces the influence of superficial LaTeX variation, a high Score@CDM but much lower ExpRate@CDM suggests that many predictions are locally close yet still fail at the whole-expression level. This is consistent with structural inconsistency, alignment errors, or global semantic breakdown, but does not uniquely identify one cause. We will soften this claim accordingly.
>
> **On the failure mode analysis.**
> We agree that the experimental basis should be stated more clearly. Our analysis was originally grounded in the representative examples in Figure 4, and we have supplemented it with additional failure examples at **ImageLink[https://bashify.io/i/WnhIQQ ]**. These examples consistently show errors in 2D structure recognition, attention drift in long expressions, contextual symbol confusion, artifact-induced mistakes, and instruction-following failures. We will make clear that this is qualitative evidence rather than a separately quantified failure-type benchmark.
>
> **On the ExpRate@CDM threshold.**
> We will explicitly state in Section 5.1 that the equivalence threshold for ExpRate@CDM is 1.0, i.e., a prediction is counted as correct only when it is exactly equivalent to the reference under CDM.
>
> **On SASR’s reported gains.**
> We will move the Score@CDM result for SASR from the appendix to the main discussion for completeness. We will also clarify that our claim is not that SASR removes the Score@CDM–ExpRate@CDM gap entirely. Rather, SASR improves the conversion from partially correct but globally inconsistent predictions to fully correct expressions, especially in categories with stronger structural demands. This is better reflected by ExpRate than by Score@CDM. Our ablations further show that the gain mainly comes from the interaction of schema constraints and prior calibration, rather than from generic extra reasoning alone.
>
> If there are still any concerns, we would be happy to further clarify them.

---

> > ### Author Rebuttal · Reviewer_PJ4H · 2026-04-02
> >
> > I thank the authors for their willingness to clarify details in their rebuttal.
> >
> > For each of my questions:
> >
> > 1. Thanks for providing the Uni-MuMER numbers. What’s the ExpRate@CDM?
> > 2. Qualitative results such as those provided would be good backing for the failure mode analysis, and I suggest the authors include it in the main manuscript. I need one clarification, though: what is the model that generated the shown predictions? This analysis must be done, as much as possible, with some specific reference to the models and, if possible, results from multiple models (possibly the strongest for the category), otherwise, it risks being an imprecise generalization. The same goes for the failure mode analysis for RCE (for which the used models would also be a useful addition), provided in answer to Reviewer JnhP.
> > 3. Thanks for clarifying this detail. From this follows the necessity, with which the authors agreed, to soften the claim made about the gap between Score@CDM and ExpRate@CDM.
> >
> > Regarding the highlighted weakness of reliance on a custom LaTeX environment, I still believe SASR is given an unfair advantage due to in-context learning that includes samples using the custom syntax, and this also could apply to other categories, especially when not using CDM. Regardless, the inclusion of CDM (which also excludes SDE and VAE) numbers mitigates this, as well as the ablation on the individual contribution of MSR and SIL.

---

> > > ### Author Response · Authors · 2026-04-03
> > >
> > > Thank you for the follow-up and for the helpful clarifications. We provide the following additional responses.
> > >
> > > 1. **On Uni-MuMER’s ExpRate@CDM.**
> > >    Uni-MuMER achieves **17.10% ExpRate@CDM**, with **37.91% / 13.22% / 15.32% / 10.54%** on **D1/D2/D3/D4**, respectively. This suggests that it retains some capability on relatively simpler single-line expressions, but still degrades substantially on dimensions involving complex 2D structures, alignment, and real-noise conditions.
> > >
> > > 2. **On the models used in the failure mode analysis.**
> > >    Thank you for this important suggestion. In the current examples shown in the paper, the cases for “unstable global 2D structure” and “local symbol correctness without global semantic validity” are from **Qwen3-VL-8B**, which we selected because it is one of the strongest overall models and its outputs are relatively representative and easy to interpret. These failure types, however, are not unique to Qwen3-VL-8B and were also observed in other models. In particular, the “locally plausible but globally invalid” pattern is also common in **Nemotron-Nano-12B-v2-VL**, while it appears less frequently in stronger models such as **Qwen3-VL-8B** and **GLM-4.6v**. For the RCE failure analysis, the shown examples are from **Qwen3-VL-8B** and **GLM-4.6v**, and the same error types were also observed in weaker-performing models, often even more frequently. We will add such qualitative results to the main manuscript with explicit model attribution.
> > >
> > > 3. **On the wording about the Score@CDM–ExpRate@CDM gap.**
> > >    Thank you for this suggestion. We will revise the wording accordingly and present this gap more cautiously in the paper.
> > >
> > > 4. **On custom LaTeX environments and in-context learning.**
> > >    We agree with the reviewer that this point should be clarified more carefully. For models that do not support such prompting (e.g., smaller models or OCR/MER systems), we exclude **SDE** and **VAE** from the comparison and from the averaged results to avoid unfairness caused by prompt controllability. For models that support custom prompting, the comparison is fair because both the baseline and SASR receive the same custom LaTeX-format instruction. We will make this evaluation protocol clearer in the manuscript to avoid misunderstanding.
> > >
> > > We hope these clarifications address the reviewer’s concerns.

---

### Decision · Program_Chairs · 2026-04-30

**Decision:**

Accept (regular)

**Comment:**

This paper proposes HMER-Bench, a benchmark for Handwritten Mathematical Expression Recognition (HMER), together with Schema-Anchored Structure-aware Reasoning (SASR), a training-free inference framework. HMER-Bench diagnoses that current VLMs perform substantially below the expectation on full-expression recognition (~26%). SASR decomposes the task into three stages (schema reasoning, schema-constrained transcription, and context-based disambiguation) and achieves notable gains on top of Qwen-8B backbone.

The reviewers generally raised the following weaknesses: (1) limited technical depth, with a methodological contribution that does not extend much beyond a prompting strategy; (2) the main contribution lies in the benchmark, while the novelty of SASR itself is limited; and (3) insufficient comparative experiments. During the rebuttal, the experiment-related concerns were largely resolved, but the method-side weaknesses remained, and the reviewers continued to hold this view after the discussion.

The AC views this as a borderline paper. As there are no clear technical flaws, the AC respects the reviewers' final judgment and recommends acceptance.